# Molecular competition induced Janus hydrogel bioelectronic interface for electroceutical modulation

Xinyu Qu[1,2], Qian Wang [ID][1] ✉, Hanjun Sun[1], Dingli Gan[1], Youliang Zhu [ID][3] ✉, Zhenhua Ni [ID][2] ✉ & Xiaochen Dong [ID][1,4] ✉

Janus hydrogel bioelectronic interfaces have long been challenged by complex fabrication procedures, poor controllability of asymmetric properties and weak interlayer bonding strength. Herein, we fabricated a Janus hydrogel with dual structural and compositional gradients in one step via Molecular Competition Induction mechanism. Unilateral UV light-driven competitive reactions between distinct monomers induce spatiotemporal progressive polymerization, facilitating heterogeneous distribution of polymer segments and gradient-structure formation. The unique configuration effectively addresses issues of weak interfacial bonding and interlayer slippage in Janus hydrogels. Associated with the programmed directional (upward) migration of adhesive groups during fabrication, the Janus hydrogel achieved a 14.6-fold disparity in interfacial adhesion. After self-assembling patterned polypyrrole conductive percolation network on adhesive side, the Janus hydrogel bioelectronic interface enables robust and efficient bidirectional bioelectrical transduction via mechanical-electrical coupling for electroceutical modulation of abdominal wall injury and electrophysiological signals acquisition. This study provides a facile and universal approach for creating bio-adaptive Janus hydrogel interfaces.

Hydrogel bioelectronic interfaces effectively bridge the physical compatibility and functional synergy gaps between biological systems and electronic devices, enabling operative mechanical coupling with tissue interfaces and high-efficient bioelectric interaction (endogenous bioelectric signal capture and exogenous electroceutical modulation therapy)[1–7]. To this end, a fundamental requirement of robust wet bioadhesive properties in hydrogels is underscored to form conformal, continuous, and stable dynamic interface coupling with biological tissues, thereby facilitating a broad spectrum of biomedical operations[8–10]. Scientists have been dedicated to reinforcing the bioadhesion strength of hydrogels through various strategies, such as

molecular engineering, network structure regulation, functional group modifications, and multi-scale mechanical adaptation[11–15]. Despite significant progress in bioadhesion optimization, most approaches prioritize solely adhesion strength enhancement unilaterally while overlooking spatial adhesion restriction[16]. Accordingly, unintended adhesion between the non-attached side surfaces of hydrogels and non-targeted surfaces frequently occurs during application, potentially triggering electronic signal distortion and a series of complications[10].

Recently, Janus hydrogels, featuring unique heterogeneous architecture with dual-sided functionality, have garnered considerable attention owing to their remarkable multifunctional integration and

[1]State Key Laboratory of Flexible Electronics (LoFE) & Institute of Advanced Materials (IAM), School of Flexible Electronics (Future Technologies), School of Physical and Mathematical Sciences, Nanjing Tech University (NanjingTech), Nanjing, China. [2]School of Physics and Key Laboratory of Quantum Materials and Devices of Ministry of Education, Southeast University, Nanjing, China. [3]State Key Laboratory of Supramolecular Structure and Materials, Institute of Theoretical Chemistry, College of Chemistry, Jilin University, Changchun, China. [4]School of Chemistry & Materials Science, Jiangsu Normal University, Xuzhou, China. ✉e-mail: chelseawq@njtech.edu.cn; youliangzhu@jlu.edu.cn; zhni@seu.edu.cn; iamxcdong@njtech.edu.cn

precise interface regulation capability[17–20]. The rational design of Janus hydrogels with programmable asymmetric adhesion properties has become the pivotal breakthrough in addressing the aforementioned challenges in conventional bioelectronic interfaces[21]. To further enhance the adhesion disparity between the dual surfaces, scientists have proposed various innovative strategies, such as external force induction methods (gravity, buoyancy, centrifugal force, electric field, magnetic field, etc.), layer-by-layer integration, and unilateral surface modification, to amplify the interfacial asymmetry[16,19]. Nevertheless, these strategies are primarily constrained by several critical issues, such as complex fabrication procedures, harsh preparation conditions, poor reproducibility/controllability, and insufficient interlayer interfacial bonding, which pose substantial barriers to large-scale production and commercial promotion. Particularly, conventional Janus hydrogels often suffer from prominent interlayer slippage and abrupt interfacial property transitions, which directly undermine their mechanical robustness and long-term functional stability. Notably, gradient-structured Janus hydrogels, a distinct subclass defined by continuous compositional, crosslinking density, or mechanical properties variations across their thickness, effectively address these limitations with unique advantages. Their gradual property transitions replace discrete interlayer boundaries to inherently eliminate interface slippage and delamination. The gradient architecture also enables efficient stress dissipation under mechanical deformation (directly mitigating the interfacial stress concentration that plagues traditional layered designs) and facilitates synergistic integration of dual-sided functionalities (avoiding performance trade-offs caused by abrupt property jumps). Yet, fabricating such gradient structures remains challenging: precise modulation over key parameters, such as composition gradient steepness, crosslinking density distribution, demands sophisticated control systems, leading to high manufacturing costs, poor reproducibility, and limited adaptability to diverse material systems. As a result, universal and scalable fabrication strategies for gradient-structured Janus hydrogels remain absent. Consequently, developing a straightforward, highly efficient, and universally applicable fabrication approach has become a critical imperative for advancing the practical deployment and scalable manufacturing of gradient Janus hydrogels.

Free radical polymerization is a class of chain-reaction chemical processes initiated by free radical active centers, which serves as an essential strategy in hydrogel synthesis[22–24]. Notably, there is a phenomenon in experiments that is often overlooked but critical, where the gelation rate varies significantly depending on the type of polymerization precursors[25–27]. This kinetic difference may arise from intrinsic molecular structural factors and extrinsic process regulations, including thermodynamic activation barriers for radical initiation, electronic effects within the monomer's chemical structure (induction and conjugation effect), the steric bulk of monomer groups (steric hindrance) and the entanglement of polymer segments, etc., which collectively direct the polymerization process and govern the temporal evolution of the crosslinked network[28]. Therefore, the approach, by cleverly leveraging the differences in gelation rates and incorporating functional monomers with varying reactivity into the hydrogel matrix, combined with precisely engineered spatiotemporal regulation of polymerization reaction thermodynamics and kinetics, is expected to construct Janus hydrogels with programmable gradient architectures.

Herein, we selected a common monomer (acrylamide, AM) and a typical vinyl zwitterion (3-[dimethyl-[2-(2-methylprop-2-enoyloxy) ethyl] azaniumyl] propane-1-sulfonate, SBMA) as model competitive polymerizable monomer. Through a Molecular Competition Induction mechanism (Fig. 1), a Janus hydrogel with gradient structure was successfully fabricated via unilateral UV-induced directional progressive polymerization. Investigations reveal a significant polymerization rate disparity between AM and SBMA, arising from fundamental variations in double-bond electron cloud density and reaction activity[29]. The

amide group ($-CONH_2$) of AM donates electrons to the double bond through the conjugate effect, creating an electron-rich double bond with a relatively high electrostatic potential, while the quaternary ammonium group ($-N^+$) of SBMA strongly withdraws electrons through the inductive effect, forming an electron-deficient double bond with a relatively low electrostatic potential. This electronic configuration leads to distinct polymerization behaviors: the free radicals, as electrophilic species, preferentially attack the double bond of AM with a higher electron density, resulting in significantly lower polymerization activation energy and markedly higher propagation rate constant ($k_p$) of AM compared to SBMA[30–32]. Additionally, the steric hindrance effect of SBMA's side chains and the solvation effect of its ionic groups further diminish its reactivity (Supplementary Fig. S1). The kinetic difference provides the crucial driving force for constructing the Janus hydrogel: under unilateral UV irradiation, free radicals diffuse top-down, initiating explosive polymerization of AM at the bottom to form PAM chains (5-8 min), while SBMA, driven by the extrusion effect from the partially formed polymer network and competitive reaction mechanism, undergoes gradual polymerization (8-30 min) and generate upward-propagating poly(3-[dimethyl-[2-(2-methylprop-2-enoyloxy) ethyl] azaniumyl] propane-1-sulfonate) (PSBMA) chains. The spatiotemporally progressive polymerization process ultimately constructs an asymmetric structure with gradient compositional distribution. In contrast to conventional fabrication methods, we exploit controlled UV irradiation to directionally supply free radicals, facilitating precise temporal regulation on the gelation timing of small molecules and zwitterions. This approach enables progressive, bottom-up polymerization of AM and SBMA with differential timing, successfully constructing an asymmetric gradient network through elaborate kinetic modulation in both spatial and temporal domains.

To further enhance the comprehensive performance of the Janus hydrogel, N-hydroxysuccinimide ester-conjugated alginate (Alg-NHS) was introduced (Supplementary Fig. S2). On one hand, it can strengthen wet adhesion by forming covalent bonds with tissues. On the other hand, increasing the system viscosity also contributes to optimizing the heterogeneous structure. Hydroxyl-rich cellulose nanocrystals (CNCs) were incorporated into the hydrogel matrix as well to provide comprehensive mechanical enhancement, ensuring adaptability to diverse biomedical applications (Supplementary Figs. S3 and S4). The upper surface of the hydrogel, densely exposed with adhesive functional groups (NHS ester, sulfonic acid, quaternary ammonium, amine, and carboxyl groups), establishes robust interfacial interactions (covalent bonds, dipole-dipole, hydrogen bonding, and electrostatic interactions) with target tissues to guarantee conformal contact and dynamic compliance (Supplementary Fig. S5). In contrast, the bottom surface, where PAM chains accumulate and mask adhesive moieties, exhibits markedly weaker adhesion, creating a striking 14.6-fold disparity in top-bottom interfacial adhesion strength in the asymmetry structure. To expand bioelectronic functionality, a hydrogel bioelectronic interface with patterned current collector was constructed via self-assembly of pyrrole (Py) and $Fe^{3+}$ on adhesive side of the Janus hydrogel. The polypyrrole (PPy) network forms an interconnected and reliable conductive percolation network via topological and physicochemical interactions, endowing the hydrogel interface with low electrochemical impedance and efficient charge injection/ storage capabilities for electrophysiological applications (Fig. 1). Collectively, the study provides a facile and scalable approach for fabricating Janus hydrogel bioelectronic interfaces with gradient structures, paving the way for large-scale production and commercialization of this asymmetric interface.

## Results
### Characterization
To elucidate the competitive gelation mechanism, a micro infrared rheometer equipped with a UV light curing module was employed to

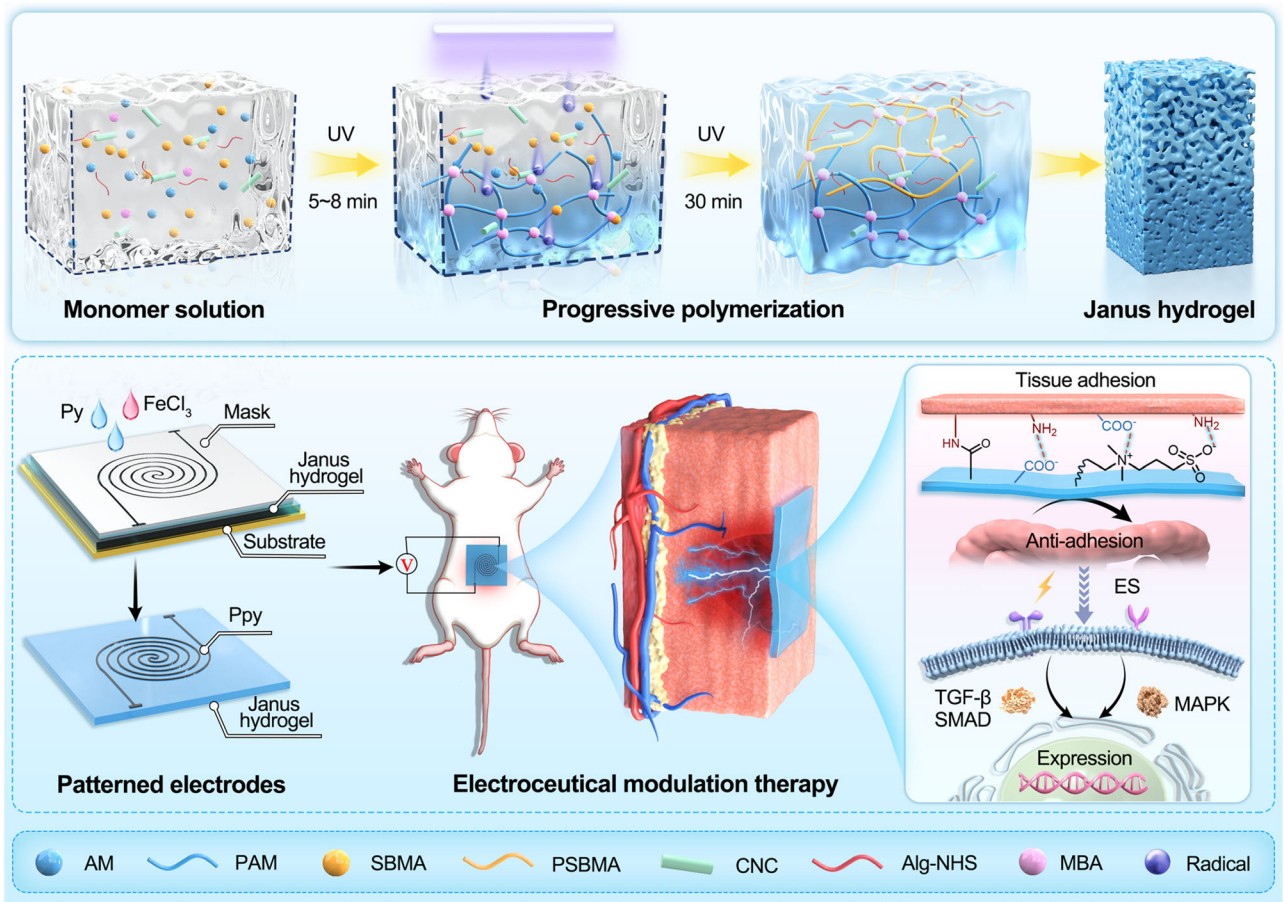

**Fig. 1 | Schematic illustration of the Janus hydrogel interface fabrication process and electroceutical modulation therapy.** Created in BioRender. Shao, J. (2025) https://BioRender.com/luci6pc.

simultaneously record real-time rheological properties and its Fourier transform infrared spectroscopy (FTIR) during the gelation process. As shown in Fig. 2a, b, the profile exhibits distinct rheological characteristics indicative of free radical polymerization. During Stage I, the (1)→(2) process represents the induction period of the free-radical polymerization, where UV irradiation drives the initiator to continuously accumulate free radicals, with no obvious infrared spectral changes. In the (2)→(4) process, the carbon-carbon double bond peak (C=C, 1601 cm$^{-1}$) of AM gradually disappears, accompanied by a significant redshift in AM's carbon-nitrogen bond peak (C−N, from 1433 to 1455 cm$^{-1}$). In comparison, the infrared absorption peaks of SBMA's carbon-hydrogen bond (=C−H, 1324 cm$^{-1}$) and sulfur-oxygen bond (S−O, 1040 cm$^{-1}$) vanish suddenly in (2)→(3) process. To verify, the gelation processes of pure PAM and pure PSBMA hydrogels were recorded (Supplementary Fig. S6). Results show that during pure PAM gelation, the infrared peaks of C=C disappearance and C−N redshift occur likewise, while for pure PSBMA gelation, =C−H peak disappears gradually, and S−O peak persists. It appears that during Stage I (336.8 s), the high concentration of free radicals boosts the rapid polymerization of AM, while the generated PAM segments settle to the bottom and displace the SBMA molecules to the upper layer, masking the featured infrared absorption peak signals. The increased relative concentration of SBMA in the upper region, with its substantial charged groups, considerably enhances intermolecular electrostatic interactions and hydrodynamic drag, markedly elevating the macroscopic viscosity and leading to a sharp rise in the loss modulus (G″) during the transition from (3) to (4). Driven by this competitive response mechanism, SBMA undergoes significant chain growth at the

onset of Stage II, revealing noticeable compositional gradient variation in the reaction system. After Stage II, the storage modulus (G′) surpasses the loss modulus (G″), the system contains no flowable precursor solution, ultimately forming a gradient-structured Janus hydrogel framework skeleton.

To systematically validate the successful fabrication of the gradient architecture, a multi-scale characterization of the hydrogel's microstructural features and physicochemical properties was conducted. Scanning electron microscope (SEM) images (Fig. 2c) reveal a well-defined gradient transition in the porous polymer network, showing progressively increasing pore density from the top (pore size 35-40 μm) to the bottom (pore size 5-15 μm). Comparatively, conventionally prepared hydrogels by thermal initiation exhibit a uniform porous structure with an average pore diameter of 25-35 μm (Supplementary Fig. S7). Furthermore, to investigate the influence of SBMA and AM content on gradient architecture formation, cross-sectional SEM images of hydrogel with insufficient SBMA and AM content were characterized (Supplementary Fig. S8). Results showed that under insufficient SBMA content, a gradient structure still formed, with pore density increasing progressively from the top (pore size 55-75 μm) to the bottom (pore size 20-25 μm). However, the insufficient SBMA led to a relative enlargement of pore size and an excessively loose network structure. In contrast, when AM content was insufficient, the gradient structure was entirely prevented (pore size 15-25 μm) owing to the absence of molecular competition-induced driving forces. To precisely map the component gradient distribution, primary amine groups (−NH$_2$) on PAM chains were fluorescently labeled and imaged via Z-Stack mode (3D reconstruction) on a Zeiss confocal microscope. The

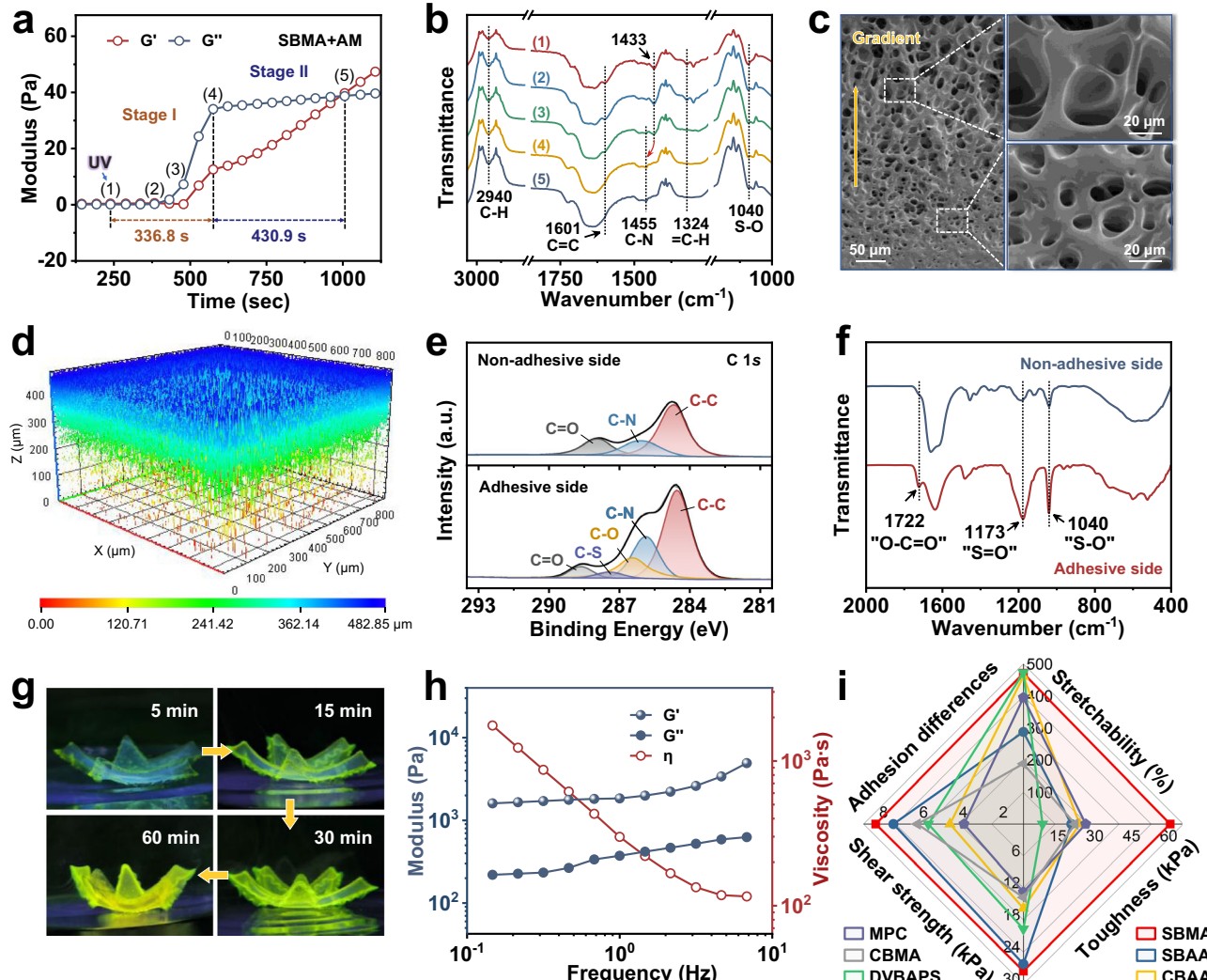

**Fig. 2 | Asymmetric features of Janus hydrogel. a, b** Rheology and corresponding FT-IR spectra of AM & SBMA during gelation progress. **c** Cross-sectional SEM images of Janus hydrogel network at different scales (*n* = 3 independent samples with similar results). **d** 3D fluorescence imaging at different locations. **e** Peak-fitting XPS spectra of the dual sides of Janus hydrogel. **f** FT-IR spectra of adhesive and non-adhesive side. **g** Behavior in water of Janus hydrogel. **h** Rheological properties of *G'* and *G''* values at different frequencies. **i** Comparison of the properties of Janus hydrogels at different zwitterionic species.

resulting 3D visualization (Fig. 2d) reveals the spatial constitution of the polymer matrix and confirms preferential PAM accumulation in the lower hydrogel layer, attributed to their faster polymerization kinetics. This directional PAM gradient also constitutes the structural basis for the Janus hydrogel's asymmetric architecture. Energy dispersive spectroscopy (EDS), X-ray photoelectron spectroscopy (XPS), and FTIR were used to characterize the surface chemical composition of the hydrogel. EDS results reveal enhanced sulfur (S) element signal (derived from PSBMA) on the top surface (Supplementary Fig. S9). XPS spectra also reveals that the top surface exhibits characteristic peaks of quaternary ammonium groups ($-N^+$) and sulfonic acid groups ($-SO_3H$) from PSBMA (Fig. 2e and Supplementary Fig. S10). For FTIR spectral analysis, it shows significantly higher $-SO_3H$ peak intensity at the top side (adhesive side) compared to the bottom side (Non-adhesive side, Fig. 2f), confirming preferential PSBMA accumulation at the top interface. To rule out the influence of CNCs and ALG-NHS on the formation of the Janus structure, FTIR spectral analysis was performed on the Janus hydrogels without CNCs/ALG-NHS (Supplementary Fig. S11). The characteristic peaks in its spectrum were consistent with those of Janus hydrogels containing CNCs/ALG-NHS, confirming CNCs and ALG-NHS do not play a decisive role in the formation of the Janus structure.

These microstructural and spectroscopic characterizations collectively verify the compositional gradient of the Janus hydrogels, wherein PAM preferentially settles to the lower regions, and PSBMA undergoes progressive polymerization and accumulates at the upper interface. Moreover, macroscopic validation of hydrophilicity and adhesion properties further confirms the successful Janus construction. Compared to the PAM-enriched bottom surface, the top surface dominated by zwitterionic PSBMA chains, demonstrates superior water-binding capacity to enable petal-like contraction within 1 h (Fig. 2g and Supplementary Fig. S12). Specifically, the adhesion tests reveal significant top-bottom surface differentiation and unidirectional adhesion: when the hydrogel's top surface (adhesive side) was applied to moist intestinal surface, no peritoneal adhesions occurred after abdominal closure (Supplementary Fig. S13). Collectively, these microscopic, spectroscopic, and macroscopic findings validate the spatiotemporal synthetic strategy for creating gradient-structured Janus hydrogels.

The mechanical and rheological properties of the Janus hydrogel were systematically evaluated. Supplementary Figs. S14a, S15a and S16 present the typical tensile/compressive stress-strain curves of the hydrogel at different compositions and water contents, demonstrating that the incorporation of Alg-NHS and CNCs substantially enhances

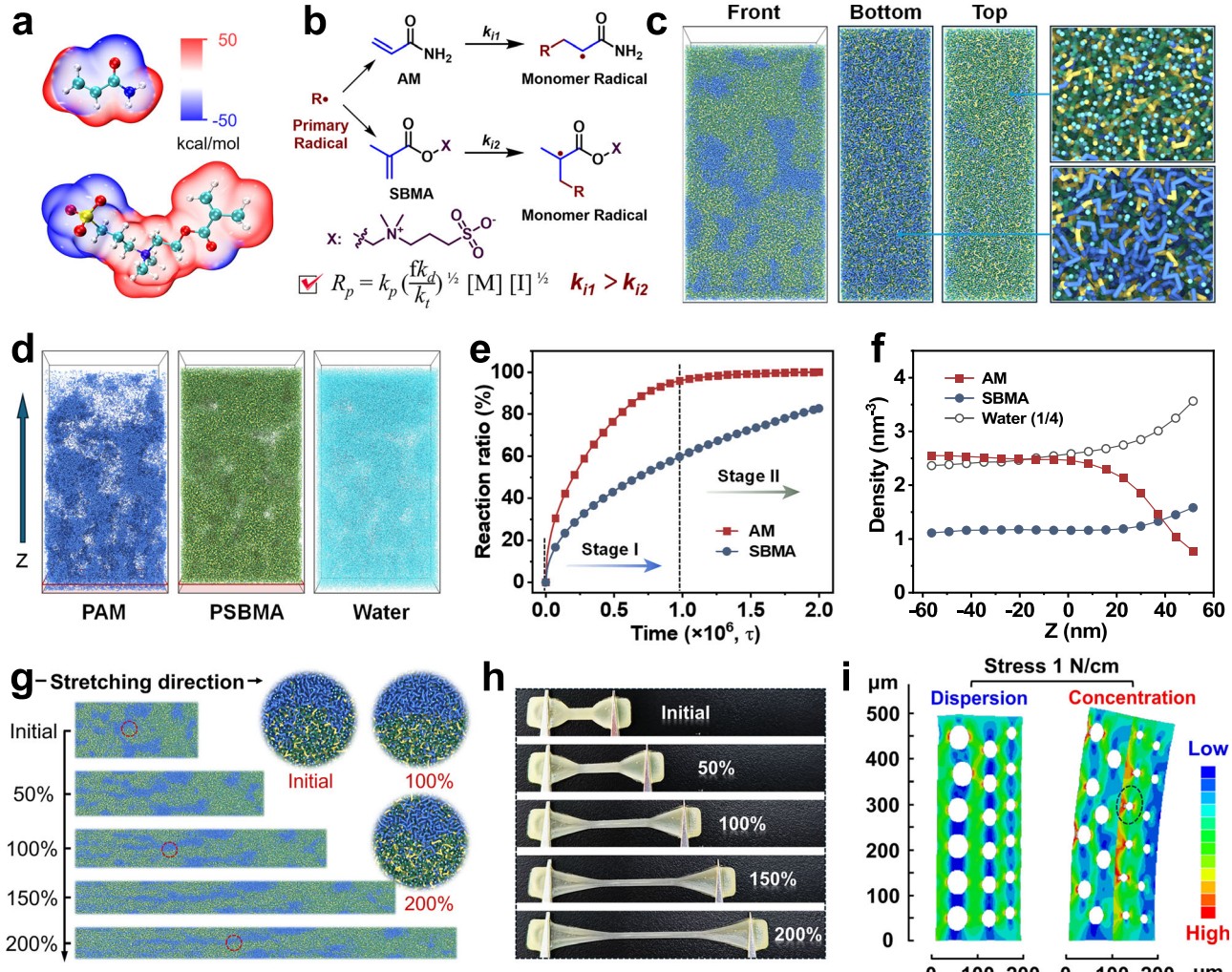

**Fig. 3 | Asymmetrical-structure formation mechanism of Janus hydrogel.**
**a** Molecular cloud density distribution of AM and SBMA. **b** Free radical polymerization kinetics. **c** Front, top, and bottom views of the Janus hydrogel model. **d** Distribution profiles of PAM, PSBMA, and water in the model. **e** Reaction ratio of AM and SBMA during polymerization. **f** Statistical analysis of asymmetric distributions of PAM, PSBMA, and water. **g** Snapshots of network evolution in the Janus hydrogel under various strains. **h** Photograph of the hydrogel state at different strains. **i** FEA of gradient structure and bilayer structure.

mechanical performance, and water content has a huge impact on mechanical properties. In Supplementary Figs. S14b, S15b, S17 and S18, the Janus hydrogel demonstrates remarkable endurance under various mechanical challenges: it rapidly recovers to its original state after successive incremental tensile/compressive strains and maintains stable performance after prolonged high-intensity mechanical loading for over 5000 s, confirming impressive mechanical resilience within physiological strain limits. Rheological characterization in Fig. 2h and Supplementary Fig. S19 reveal that the storage modulus ($G'$) consistently exceeds the loss modulus ($G''$) across a broad spectrum of testing conditions of frequency sweeps (0.1-10 Hz), temperature variations (25-45 °C) and strain amplitudes (0-100%), indicating stable viscoelastic behavior under dynamic conditions mimicking biological tissue motion. Collectively, despite its intentionally engineered asymmetric architecture, the Janus hydrogel exhibits exceptional mechanical toughness and SBMA content-dependent mechanical strength enhancement (Supplementary Fig. S20), benefiting from the interpenetrating topological entanglement enabled by the one-step fabrication process. Furthermore, the successful preparation of Janus hydrogels using combinations of various small molecules and vinyl zwitterion monomers has clearly verified the broad applicability and reliability of the strategy (Fig. 2i, and Supplementary Figs. S21, S22).

This cross-system consistency not only highlights the versatility of our spatiotemporal kinetic modulation approach but also demonstrates its potential for scalable and reproducible synthesis across different material systems.

## Asymmetrical-structure formation mechanism

Influenced by the electron cloud density and reaction activity of double bonds, as well as the steric hindrance of side chains, the polymerization kinetics of AM and SBMA exhibit significant differences (Fig. 3a, b)[33,34]. To elucidate the fundamental mechanism underlying gradient structure formation and the directional arrangement of polymer segments in hydrogels under unilateral UV irradiation, the coarse-grained molecular dynamics (CG-MD) simulation was employed to achieve a large temporal and spatial scale for complex polymerization process (Supplementary Fig. S23). Specifically, a monomer of AM was coarse-grained into a single *Na* bead, and a monomer of SBMA was represented by the three connected beads of *Na1-QO-Qa* named from left to right. A cluster of four water molecules was represented by a *P4* bead. The bead-spring model with Martini force field methodology was employed in this CG-MD study[35]. The free radical polymerization of AM and SBMA monomers in aqueous medium generated polyacrylamide (PAM) and PSBMA chains. To simulate

the unilateral UV effect, radicals were introduced from the positive $Z$-axis direction. As the radical concentration progressively increased and exceeded the kinetic activation barrier, the chain polymerization of AM monomers was triggered, driving rapid PAM chain formation. The growing PAM chains underwent bottom-phase deposition due to limited solubility and gravitational induction effect (blue chains in Fig. 3c). Concurrently, SBMA experienced kinetically restricted chain polymerization (likely from steric hindrance or chain transfer effects), resulting in extended PSBMA chains that permeated the network matrix. After complete consumption of AM monomers, the remaining SBMA monomers continued polymerizing to establish a distinct PSBMA-rich layer at the top surface (yellow-green polymer chains in Fig. 3c), ultimately accomplishing the gradient asymmetric architecture (Supplementary Fig. S24).

Figure 3e quantitatively illustrates the temporal consumption profiles of AM and SBMA monomers. During Stage I, AM monomers underwent rapid depletion with a significantly higher reaction ratio than SBMA, indicating higher reaction kinetics. In Stage II, AM monomers became nearly exhausted and absent from participating in spatial network construction, while SBMA maintained a constant reaction rate throughout the designated timeframe. This differential polymerization kinetics, arising from AM's explosive polymerization versus SBMA's moderate reaction kinetic, led to a $Z$-axis gradient distribution of PAM chains within the hydrogel network, contrasting sharply with the relatively uniform PSBMA chains distribution (Fig. 3d). Density field analysis further corroborated this structural asymmetry, showing decreased PAM density at the top region, accompanied by increased PSBMA density (Fig. 3f). These computational results conclusively demonstrate the formation mechanism of asymmetric hydrogels, which are in exceptional consistency with experimental observations.

To underscore the superiority of our gradient hydrogels over conventional bilayer hydrogels, in silico visualization of network evolution was performed under 0-200% strain (Fig. 3g). Using an iso-volumic tensile test mode, the box length in the $Z$-direction increased linearly, while the $X$- and $Y$-direction lengths contracted proportionally to maintain constant volume. As exhibited, the gradient-structured PAM/PSBMA network forms interpenetrating topological entanglement, where hydrogen-bonded PAM segments (blue) and ionic PSBMA domains (yellow-green) dynamically interweave. Unlike bilayer hydrogels, which suffered from interfacial delamination, the synergistic architecture dissipates energy through sequential hydrogen bond dissociation, maintaining structural integrity even under large tensile deformations (Fig. 3h and Supplementary Fig. S25). Furthermore, finite element analysis (FEA) further demonstrates that the gradient-structured Janus hydrogel rapidly dissipates stress when subjected to bending and stretching, thereby effectively mitigating interfacial stress concentration and interfacial slippage (Fig. 3i and Supplementary Figs. S26, S27).

## Asymmetric adhesion performance

During the polymerization process, competitive aggregation of PAM segments at the bottom sterically hinders adhesive group exposure while simultaneously inducing directional migration of these adhesive moieties. This spatial progressive polymerization further drives a distinct $Z$-axis compositional gradient within the polymer network, ultimately generating a significant top-bottom interfacial adhesion strength disparity. To systematically evaluate the asymmetric adhesion properties of Janus hydrogel, we employed a multi-modal testing method, including burst pressure, lap shear, and 90° peel tests. All tissue substrates used for subsequent tests were maintained in a humid state to accurately simulate real physiological conditions. First, the sealing and conformal capacity of Janus hydrogels on wet tissues was evaluated using a burst pressure measurement. In Fig. 4a, b, a custom-designed testing device with precisely tunable air pressure was established, where a pigskin defect was sealed with Janus hydrogel.

After interfacial wet bonding, the air pressure was incrementally increased until the seal failure, while recording the critical pressure at the interface (termed burst pressure) (Supplementary Movie 1). As shown in Fig. 4c, the Janus hydrogel exhibits distinct adhesion asymmetry between its two interfaces. With increasing SBMA content, both the adhesive and non-adhesive sides demonstrate enhanced adhesive strength. The Janus hydrogel achieves an optimal asymmetric adhesion performance at 37.5 wt% SBMA content. At this critical point, the adhesive side exhibits a burst pressure of 118.9 mmHg, exceeding commercially available professional bio-adhesives (Vetbond®, Tegaderm®, and fibrin glue), while exhibiting a 7.3-fold enhancement over the non-adhesive side[36,37].

Subsequently, the lap shear and 90° peel tests were conducted to evaluate the interfacial bonding strength of Janus hydrogel (Fig. 4d–g). Specifically, the Janus hydrogel was firmly pressed onto diverse biological tissue substrates (heart, intestine, liver, muscle, and skin) and engineered solids (Au, Cu, glass, and polyimide) to establish stable adhesive interfaces. Following standardized testing protocols, the lap shear and 90° peel measurements were systematically performed. As shown in Fig. 4h, i, l, and Supplementary Fig. S28, the adhesive side exhibits superior shear strength (up to 40.17 kPa) and interfacial toughness (up to 110.48 J·m$^{-2}$) across all tested substrates, achieving a maximum of 14.6-fold enhancement compared to the non-adhesive side. Meanwhile, the effects of CNCs/ALG-NHS and water content on asymmetric adhesive performance were simultaneously investigated. Results demonstrated that the incorporation of CNCs/ALG-NHS enhanced the adhesive performance and further widened the adhesion disparity between the two sides (Supplementary Fig. S29a). Moreover, reducing the water content increased the adhesion strength on both sides, with the optimal asymmetric adhesion observed at 70% water content (Supplementary Fig. S29b). This pronounced interfacial asymmetry of Janus hydrogel in tissue adhesion strength highlights the unique functionality of a heterogeneous biointerface. Furthermore, the adhesive side maintains high shear strength and interfacial toughness throughout multiple adhesion-detachment cycles and extended substrate stretching, revealing exceptional durability and operational reliability (Fig. 4j, k).

## Electrical/electrochemical behaviors

To achieve bioelectronic functionality, the adhesive side of the Janus hydrogel was pre-patterned through a waterproof mask and treated with Py and Fe$^{3+}$ solutions to construct a customized conductive percolation network (Supplementary Fig. S30). The continuous PPy networks endow the hydrogel interface with a superior conductivity exceeding 8.7 S/m, which is an order of magnitude higher than biological tissues (<0.7 S/m) (Fig. 5a)[8]. Remarkably, the hydrogel interface retains high conductivity (>7.6 S/m) after 3000 stretching cycles, demonstrating robust durability and reliability (Supplementary Fig. S31). For electrochemical performance, as shown in Fig. 5b, and Supplementary Figs. S32, S33, the PPy patterned Janus hydrogel interface exhibits higher current density and maintains stability after 1000 charge-discharge cycles and long-term PBS solution immersion in cyclic voltammetry (CV) tests, demonstrating excellent charge storage capability (CSC) and efficient electron transfer with minimal energy loss. Figure 5c and Supplementary Fig. S34 show electrochemical impedance spectroscopy (EIS) of the hydrogel interface, wherein it exhibits a relatively low interfacial impedance (<100 Ω) across the $10^{-1}$-$10^4$ Hz frequency range and remains stable after 1000 stretching cycles. Additionally, following 7 days of immersion in PBS, the interfacial impedance remained within a favorable and acceptable range, exhibiting a minor increase attributed to the slight attenuation in adhesive performance (Supplementary Fig. S35). Under ±0.5 V biphasic pulse stimulation, the charge injection capacity (CIC) value of the hydrogel interface is significantly higher than the control group and retains negligible decay (<2.2% loss) after 3000 bipolar

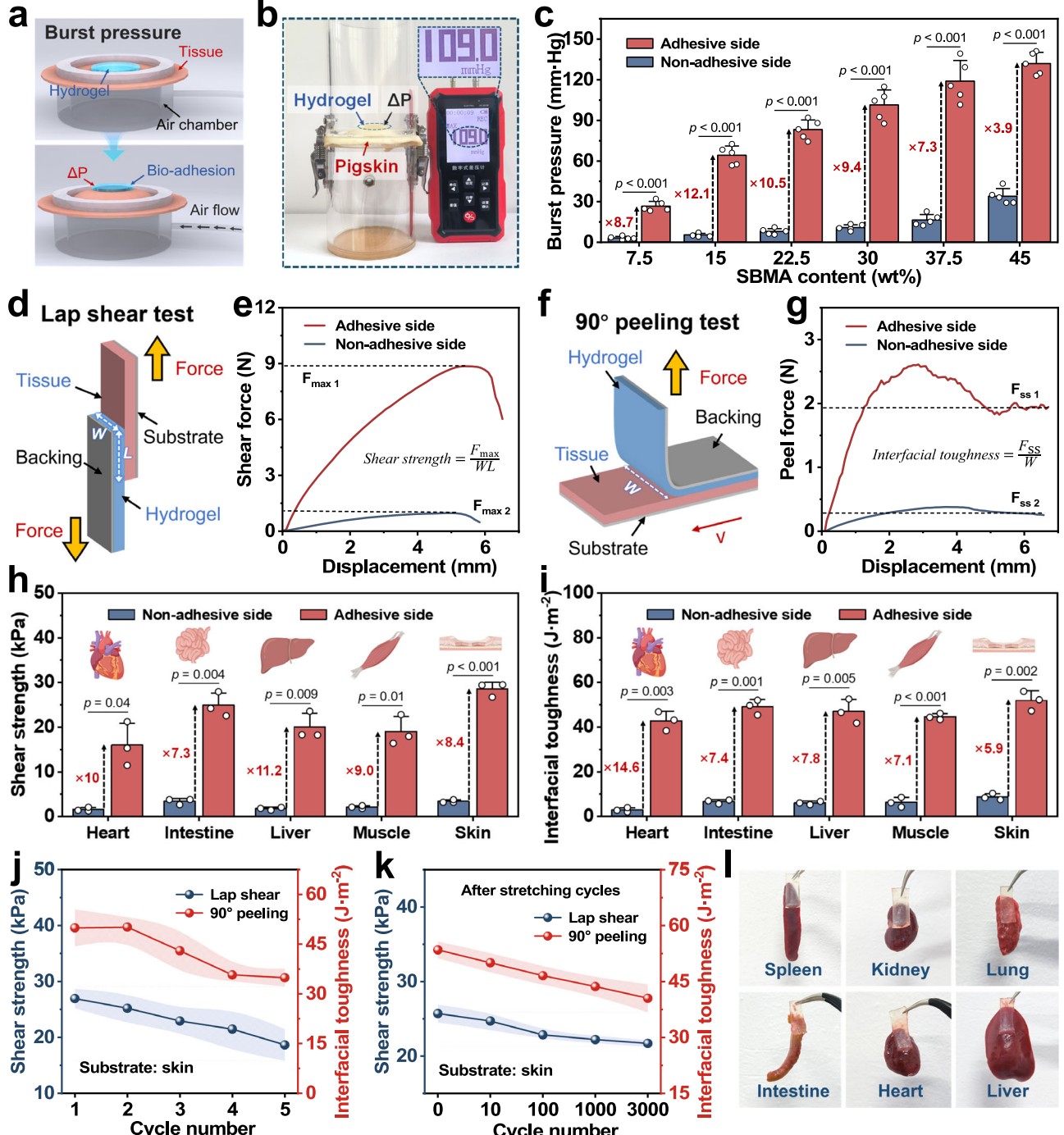

**Fig. 4 | Asymmetric adhesion performance of Janus hydrogel. a, b** Illustrations and photographs of burst pressure tests. **c** Burst pressure curves at different SBMA contents ($n$ = 5 independent samples). **d–g** Schematic illustrations and force curve of lap shear and 90° peel tests (substrate: skin). **h, i** Shear strength and interfacial toughness with different biological tissues ($n$ = 3 independent samples). **j, k** Adhesion performance of the adhesive side at varied adhesion cycles and substrate stretching at 25% strain. **l** Images of the Janus hydrogel adhered to biological tissues. **h, i** Created in BioRender. Shao, J. (2025) https://BioRender.com/luci6pc.

charge injections, demonstrating superior charge injection stability (Fig. 5d and Supplementary Fig. S36). Given the practical demand for low-voltage stimulation, the CIC of the hydrogel interface was further tested under ±0.05 V biphasic pulse stimulation (Supplementary Fig. S37). The hydrogel interface demonstrated robust long-term stability and efficient CIC performance under this low stimulation voltage, validating its adaptability to the low-voltage requirements of clinical bioelectronic stimulation (Supplementary Fig. S38). As shown in Fig. 5e, under 1 Hz/0.1 V sinusoidal voltage input, the Janus hydrogel

interface generates 32-fold higher response current than the PPy-free hydrogel and retains 99.1% signal fidelity after 1000 cycles, confirming its reliability as an electrical stimulation medium. Additionally, response currents were recorded for different frequencies of sinusoidal voltage inputs (Fig. 5f and Supplementary Fig. S39). Strikingly, the PPy-free hydrogel suffers 83.9% current drop and severe waveform distortion at low frequencies (0.1 Hz), whereas the Janus interface maintains near-sinusoidal responses across 0.1-50 Hz, demonstrating superior dynamic response features.

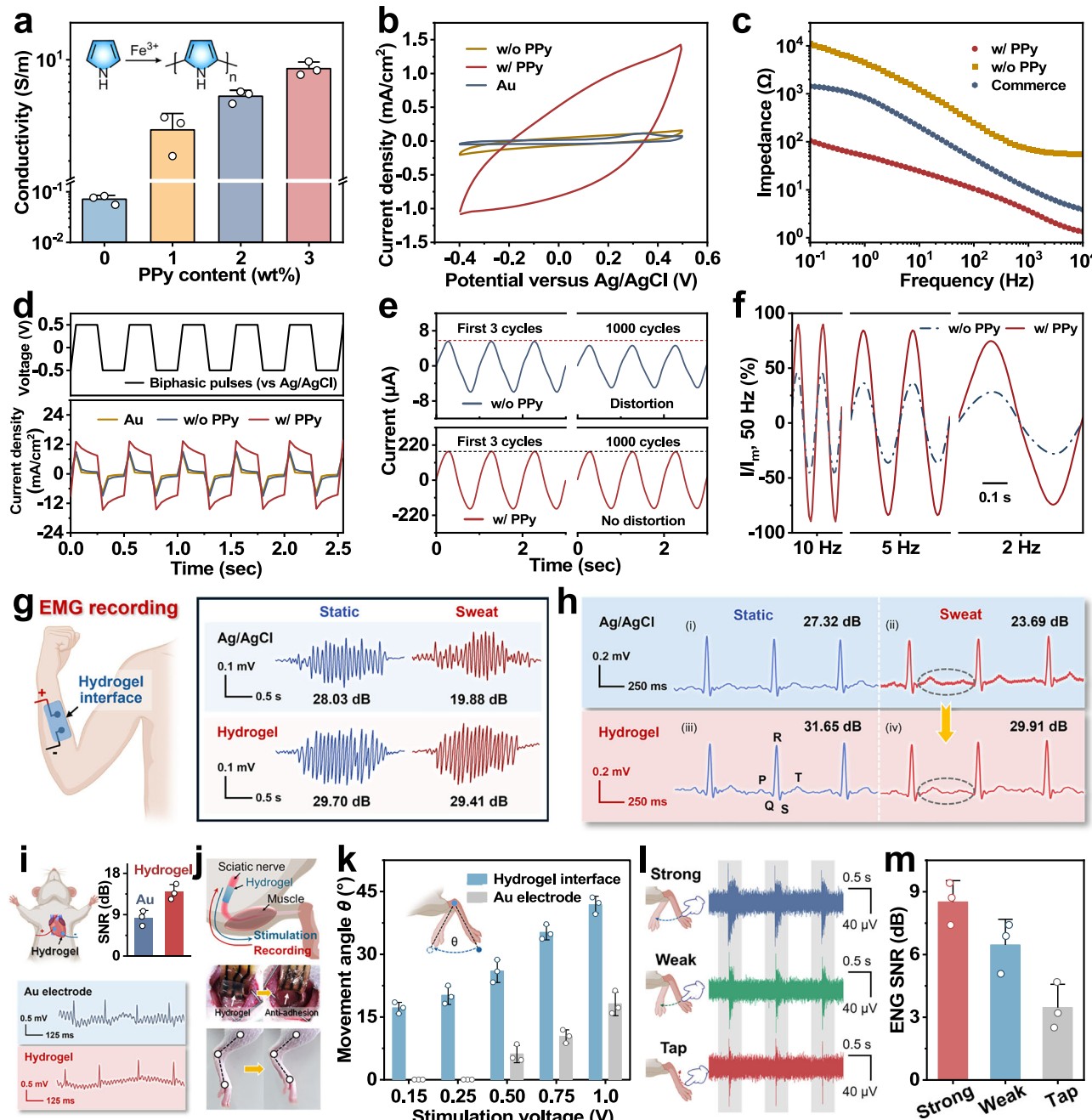

**Fig. 5 | Electrical/electrochemical behaviors of Janus hydrogel interface.**
**a** Conductivity at different PPy contents ($n = 3$ independent samples). **b** CV curves.
**c** Interfacial impedance. **d** Charge injection curves with biphasic pulses of ±0.5 V.
**e** Current *vs.* time curves upon sinusoidal AC voltage (amplitude 0.1 V, frequency
1 Hz). **f** Normalized current *vs.* frequency curves upon sinusoidal AC voltage.
**g**, **h** EMG and ECG signals captured by commercial Ag/AgCl electrode and Janus
hydrogel interface at static and sweat states. **i** Epicardial ECG and its SNR captured
by Au electrode and Janus hydrogel interface ($n = 3$ independent samples).

**j** Schematic illustration and photograph of Janus hydrogel interface onto the sciatic
nerve for electrical stimulation and ENG recording. **k** Joint angle changes triggered
via Janus hydrogel interface and Au electrode at different stimulation voltages
($n = 3$ independent samples). **l**, **m** ENG signal and its SNR of the sciatic nerve
triggered by mechanical stimulation (strong beat, weak beat, and light tap) ($n = 3$
independent samples). **g**, **i**–**l** Created in BioRender. Shao, J. (2025) https://
BioRender.com/luci6pc.

The exceptional electrical and electrochemical properties of the
Janus hydrogel interface lay a solid foundation for bioelectronic
applications. As demonstrated in Fig. 5g, h, and Supplementary
Fig. S40, the hydrogel interface exhibits superior capability for elec-
trophysiological signal acquisition (Electromyogram, EMG; Electro-
cardiogram ECG; Electrooculogram, EOG). In EMG monitoring under
fist-clenching movement, the hydrogel interface distinctly captured
the signal, achieving a signal-to-noise ratio (SNR) of 29.70 dB, com-
parable to commercial Ag/AgCl gel electrodes (28.03 dB). Notably,

under sweaty conditions, it robustly maintained high signal accuracy
with a SNR of 29.41 dB, whereas the commercial Ag/AgCl gel electrodes
declined considerably to 19.88 dB. For ECG monitoring, the hydrogel
interface clearly recorded PQRST waveforms with an SNR of 31.65 dB,
slightly surpassing the commercial electrodes (27.32 dB). When
exposed to sweat, its SNR remained as high as 29.91 dB, far exceeding
the commercial baseline (23.69 dB). In EOG monitoring, the hydrogel
electrode also accurately distinguished directional eye movement with
SNRs of 16.32 dB (leftward) and 17.50 dB (rightward), outperforming

commercial Ag/AgCl gel electrodes (11.52 dB and 10.80 dB, respectively; Supplementary Fig. S40). Integrated with its tissue-compatible adaptive asymmetric adhesion and mechanical compliance, the hydrogel interface exhibits significantly enhanced anti-interference and anti-wetting capabilities under sweat and dynamic movements, highlighting its potential for high-precision monitoring of human electrophysiological signals and electrophysiology-based control applications.

To further explore the potential utility of the Janus hydrogel interface in implantable settings, we conducted in vivo electrophysiological recording and stimulation in a rat model. For in vivo ECG monitoring, the Janus hydrogel interface was implanted on the epicardial surface of rats (Fig. 5i). Owing to its superior electrochemical properties and wet adhesion capability, the Janus hydrogel interface enabled continuous and dynamic ECG acquisition during rapid and vigorous cardiac motion, with a SNR as high as 13.9 dB, significantly higher than that of commercial gold (Au) electrodes (8.3 dB). To further verify the capability of electroneurogram (ENG) signal acquisition and low-voltage stimulation, the Janus hydrogel interface with refined multi-functional sites was affixed to the sciatic nerve of rats (Fig. 5j). Notably, the non-adhesive backside of the Janus hydrogel interface exhibited excellent anti-adhesion properties, effectively minimizing implantation-associated issues (e.g., inflammatory responses and unwanted tissue adhesion) and motion artifacts and improving SNR. Leveraging the refined multi-functional sites, the electrode enables flexible acquisition of multi-site neural signals acquisition and delivery of localized electrical stimulation, demonstrating broad adaptability across diverse application scenarios ranging from basic research to potential clinical settings. For neural low-voltage stimulation, the Janus hydrogel interface elicited leg joint movement in rats at a lower threshold voltage of 0.15 V, which was substantially superior to commercial Au electrodes (0.5 V), demonstrating excellent neuromuscular activation efficacy (Fig. 5k). Furthermore, under low-frequency electrical stimulation (<30 Hz), the Janus hydrogel interface achieved a 100% success rate in triggering leg movement, confirming its robust neuromodulatory capability (Supplementary Fig. S41). For high-fidelity electrophysiological signal recording, the hydrogel interfaces successfully captured ENG signals with high SNR following various mechanical stimulations, including strong beat, weak beat, and light tap on the rat foot (Fig. 5l, m). This superior performance fully exploits the structural advantage of the Janus design: during leg movement, the adhesive side maintains tight adherence to the sciatic nerve to establish a reliable conductive connection, while its non-adhesive side preserves lubrication with surrounding tissues, avoiding signal interference caused by unwanted adhesion and thus ensuring consistent signal quality. Benefiting from its excellent electrical/electrochemical properties and asymmetric structural advantages, the Janus hydrogel interface enabled both low-voltage neural stimulation and high-fidelity electrophysiological signal recording, providing a robust platform for efficient and precise neural modulation.

## Electroceutical modulation for abdominal wall reconstruction

Abdominal wall defects represent a clinical challenge characterized by delayed healing kinetics and high complication rates, including persistent inflammation, recurrent abscess formation, and severe visceral adhesion-associated morbidities. The Janus hydrogel bioelectronic interface, featuring unique asymmetric adhesion, tissue-like modulus, and excellent electrochemical capability, can closely adhere to abdominal wall defects, effectively resisting abdominal pressure and preventing tissue adhesion. Additionally, the hydrogel interface can induce directed cell migration and differentiation via intentional electroceutical modulation, thereby accelerating the abdominal wall reconstruction process. To ensure the safety of electroceutical modulation, flow cytometry was employed to analyze cell viability after electrical stimulation (ES) treatment (Supplementary Fig. S42). Results

demonstrated high cell viability across all ES-treated groups (94.9% viability in 100 mV/mm group), confirming the selected ES parameters do not induce significant cell death and are biocompatible for in vitro cell experiments.

To ensure long-term implantable biosafety and efficacy, the cytotoxicity, hemocompatibility, host response and key electrochemical parameters of Janus hydrogel were evaluated. As shown in Supplementary Figs. S43–S45, mouse fibroblasts (L929) were employed to assess the biocompatibility of the hydrogel using MTT assay, live/dead cell staining assay, and flow cytometry. After 24-h co-incubation with the hydrogel and its extracts, no significant cell death was observed, demonstrating negligible cytotoxicity. Hemolysis assays further confirmed the excellent hemocompatibility of the hydrogel (Supplementary Fig. S46). Compared to the control group, host response evaluation following the hydrogel implantation reveals negligible differences in hematological/biochemical parameters, and histological structures of major visceral organs (heart, liver, spleen, lung, kidney) (Supplementary Figs. S47 and S48). Furthermore, the Janus hydrogel interface was implanted subcutaneously to investigate the stability of its long-term electrochemical properties in vivo. It is identified that CIC and CSC values remained highly stable throughout the implantation period, which confirms the superior charge injection stability and long-term functional reliability of the PPy patterned Janus hydrogel interface for chronic bioelectronic applications (Supplementary Fig. S49).

To evaluate the wound healing promoting effect of electrical stimulation delivered through the Janus hydrogel bioelectronic interface, we conducted an in vitro scratch wound assay to simulate the wound healing process (Fig. 6a). The results reveal significantly enhanced L929 cell proliferation and migration efficiency in the electrical stimulation group (Fig. 6b). As shown in Supplementary Fig. S50, the initial relative wound area is ~58.1%. After 24 h of culture, the electrical stimulation group achieves a dramatic wound closure of 4.2%, representing a nearly 6.5-fold improvement in healing efficiency compared to the control group (27.1%). Additionally, the in vitro proliferative behavior of L929 cells under electrical stimulation was independently investigated, and a cell counting kit-8 (CCK-8) assay was used to quantify cell proliferation rates. As shown in Fig. 6c and Supplementary Fig. S51, a time-dependent positive correlation is observed in absorbance ratios of the hydrogel + ES group (relative to the control group), indicating significantly enhanced cell viability and proliferation following electrical stimulation. To validate the proliferative effect of electrical stimulation, flow cytometry analysis of fluorescein isothiocyanate (FITC)-labeled L929 cells was employed (Fig. 6d). Results reveal higher fluorescence intensity in the electrical stimulation group, confirming the positive impact of hydrogel-mediated electrical stimulation on cell proliferation.

To validate its therapeutic effectiveness in vivo, we surgically created a rat abdominal wall defect model involving defects in the lateral abdominal wall spanning the external oblique, internal oblique, and transversus abdominis muscle planes. Subsequently, the hydrogel interface was closely apposed and sealed onto the wound defect with the adhesive side, followed by cyclic electrical stimulation therapy (Fig. 6e). Notably, the specifically designed tree-ring-like spiral patterned hydrogel interface enables precise conformity to irregular boundaries of the multi-layer muscle defect, uniform radial distribution of the electric field across the defect area, and differential regulation of stimulation intensity between the defect center and edge, ensuring efficient, targeted delivery of electrical signals to repair-related cells in the muscle layers and providing a baseline for organized tissue regeneration (Supplementary Fig. S30a). To evaluate remodeling efficiency, pathological assessments of the traumatized abdominal wall tissues were performed using hematoxylin-eosin (H&E) and Masson trichrome (Masson) staining (Fig. 6f, g). It is identified that the electrical stimulation therapy group exhibits increased collagen

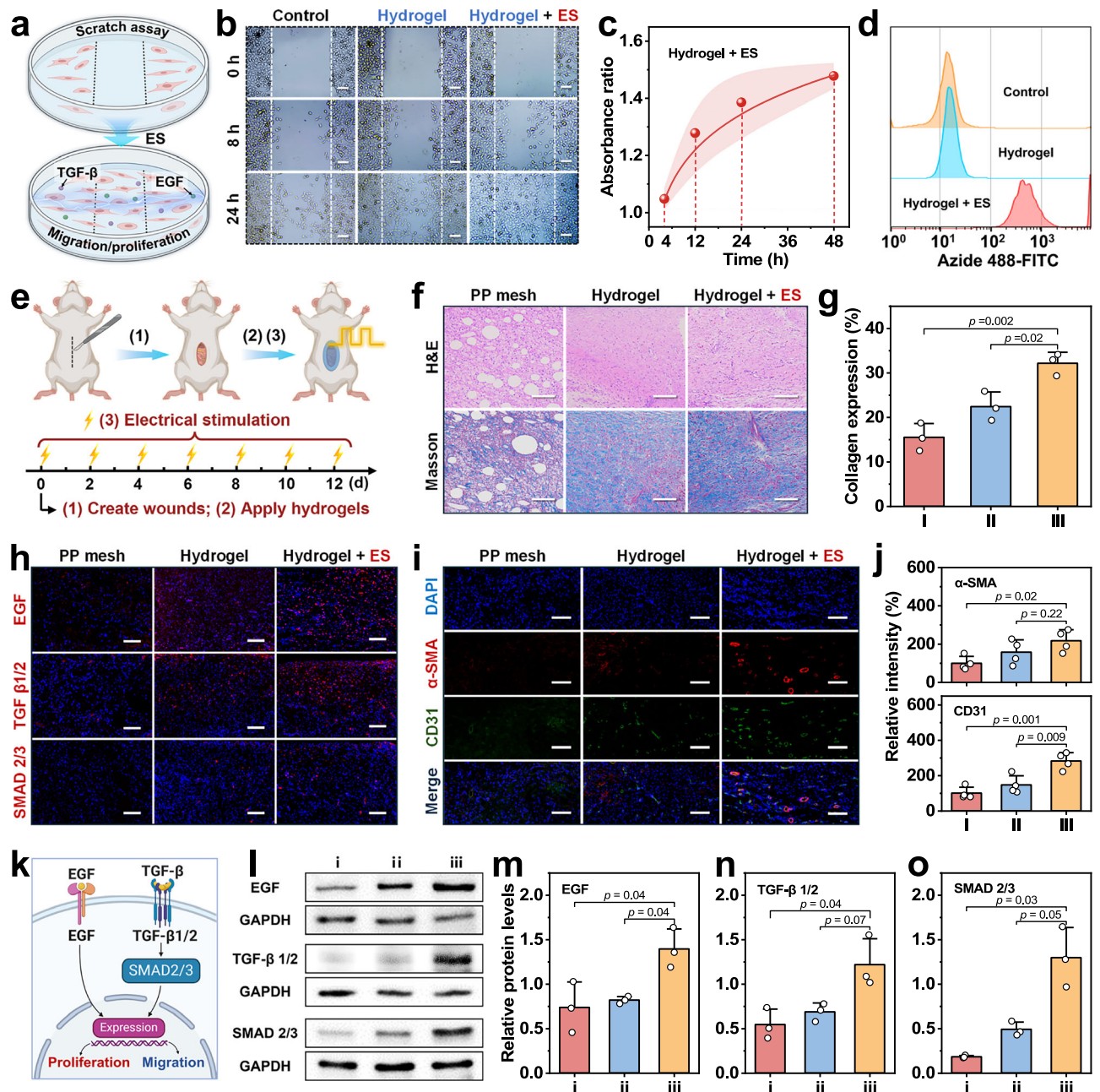

**Fig. 6 | Electroceutical modulation via Janus hydrogel interface. a** Schematic of electrical stimulation promoting cell migration and proliferation. **b** Microscopic images of the scratch assay (*n* = 3 independent samples with similar results). Scale bar: 50 μm. **c** Absorbance ratio of cells in the hydrogel + ES group to those in the control group by CCK-8 assay. Color bars indicate standard deviations. **d** Flow cytometry results of electrical stimulation therapy group and control group. **e** Schematic illustration of electrical stimulation therapy. **f, g** Histological analysis on day 21 (*n* = 3 independent samples). **h** Representative fluorescence images of

EGF (red), TNF-*α* (red) and SMAD 2/3 (red) in the tissues. Scale bar: 100 μm. **i** Representative fluorescence images of *α*-SMA (red) and CD31 (green) in the tissues. Scale bar: 100 μm. **j** Fluorescence intensity in the tissues (*n* = 3 independent samples). **k** Mechanism of electrical stimulation to accelerate wound healing. **l** Western blot detection and quantification of EGF and TGF-*β* (n = 3 independent samples with similar results). **m**–**o** Relative protein levels of EGF, TNF-*α*, and SMAD 2/3 (*n* = 3 independent samples). **a, e, k** Created in BioRender. Shao, J. (2025) https://BioRender.com/luci6pc.

deposition and angiogenesis, indicating superior tissue remodeling. Complementary immunofluorescence staining for CD31/*α*-smooth muscle actin (*α*-SMA) was employed for quantitatively mapping vascular maturation. Comparatively, the electrical stimulation therapy group shows significantly enhanced fluorescence intensity in angiogenic regions, demonstrating advanced blood vessel formation during healing (Fig. 6i, j). Additionally, immunofluorescence staining for the pro-inflammatory cytokine TNF-*α* reveals reduced inflammatory infiltrates in the hydrogel-covered group, demonstrating the hydrogel's

capacity to suppress inflammatory intensity and accelerate resolution of the inflammatory peak (Supplementary Fig. S52). This anti-inflammatory effect is fundamentally linked to the Janus hydrogel's unique asymmetric adhesion. Specifically, the hydrogel's adhesive side forms a tight, conformal seal over the abdominal wall defect, which not only facilitates rapid wound closure to minimize exposure to pro-inflammatory stimuli but also anchors the hydrogel securely against abdominal pressure. Concurrently, the non-adhesive side delivers exceptional anti-adhesion performance, serving as a critical physical

barrier against visceral adhesion and bacterial contamination in the abdominal cavity, both of which are major triggers of persistent inflammation. By simultaneously ensuring secure adhesion and offering anti-adhesion protection, the Janus structure effectively eliminates these exacerbating factors, thereby reinforcing its overall anti-inflammatory efficacy.

To elucidate the mechanisms underlying accelerated abdominal wall repair via electrical stimulation, immunofluorescence staining and western blotting were applied to track and quantify the expression of wound-healing-related growth factors and pathway proteins, including epidermal growth factor (EGF), transforming growth factor-$\beta$ (TGF-$\beta$) and mothers against decapentaplegic homolog 2/3 (SMAD 2/3). Immunofluorescence staining results reveal that there are significantly evaluated expression levels of these protein secretions in electrical stimulation group (Fig. 6h and Supplementary Figs. S53–55). Western blotting results show that there is a significant upregulation of these proteins in electrical stimulation group (Fig. 6i–o). Collectively, the Janus hydrogel interface-mediated electrical simulation activates MAPK and TGF-β/SMAD signaling cascade (Fig. 6k), which effectively accelerates cell proliferation and promotes abdominal wall reconstruction. Moreover, the coordinated activation of these pathways can simultaneously prevent tissue adhesion and reduce various postoperative complications effectively.

The integration of in vivo therapeutic outcomes and mechanistic insights above clarifies a precise "electrical stimulation-input → signaling pathway-activation → wound repair-output" regulatory axis, with the Mitogen-Activated Protein Kinase (MAPK) and TGF-$\beta$/SMAD cascades as the core molecular drivers of the Janus hydrogel system's efficacy. The key phenotypic improvements in the rat abdominal wall defect model, enhanced collagen deposition, advanced vascular maturation, and attenuated inflammation, are directly mediated by electrical stimulation-induced activation of these two pathways. Mechanistically, electrical stimulation upregulates EGF to trigger the MAPK pathway: this not only promotes proliferation of repair-related cells (keratinocytes, fibroblasts, vascular endothelial cells) to support re-epithelialization and vascular formation, but also suppresses TNF-$\alpha$ to resolve excessive inflammatory responses. Concurrently, it elevates TGF-$\beta$ and its downstream effector SMAD2/3 to activate the TGF-$\beta$/SMAD pathway, which drives fibroblast-to-myofibroblast differentiation and ordered collagen synthesis. Both processes are critical for restoring the mechanical integrity of the multi-layered abdominal wall defect. Beyond accelerating wound closure, the synergistic activation of MAPK and TGF-$\beta$/SMAD avoids disorganized extracellular matrix deposition (preventing postoperative tissue adhesion) and mitigates TNF-$\alpha$-associated inflammatory complications, addressing major clinical challenges in abdominal wall repair.

## Discussion

In summary, we proposed a Molecular Competition Induction mechanism to fabricate Janus hydrogels with gradient structures via unilateral UV-induced directional progressive polymerization of two reactive monomers. The Janus hydrogel exhibited pronounced asymmetric properties in chemical composition, morphological features, and mechanical properties. After patterning PPy percolation conductive networks on adhesive side, the hydrogel bioelectronic interface demonstrated favorable electrochemical impedance and charge injection/storage capacities for electrophysiological applications. In vivo evaluation using abdominal wall defect repair models showed that the hydrogel interface effectively prevented postoperative tissue adhesion and accelerated abdominal wall reconstruction through electrical stimulation-induced directional cell migration and differentiation. The high-efficient and standardized fabrication strategy provides significant potential for the large-scale production of Janus hydrogels, and promotes their wide-ranging commercialization in bioelectronics.

## Methods

### Materials

2-($N$-morpholino) ethane sulphonic acid (MES), $N$-hydroxysuccinimide (NHS), 1-(3-dimethylaminopropyl)-3-ethylcarbodiimide hydrochloride (EDC), α-cellulose, 3-[dimethyl-[2-(2-methylprop-2-enoyloxy) ethyl] azaniumyl] propane-1-sulfonate (SBMA), $N$, $N'$-methylene bisacrylamide (MBA), Pyrrole (Py), 2-Hydroxy-4'-(2-hydroxyethoxy)-2- methylpropiophenone (Irgacure 2959), and FeCl$_3$ were purchased from Adamas-Beta. Sodium alginate (Alg) was obtained from Sigma-Aldrich. Acrylamide (AM) was obtained from Greagent.

### Material synthesis and characterization

**Synthesis of Alg-NHS and CNCs.** For Alg-NHS, 2.16 g Alg was dissolved in 50 mL MES buffer solution (0.5 mol·L$^{-1}$). Subsequently, 0.575 g NHS and 0.75 g EDC were successively added to the mixed solution under N$_2$ atmosphere and stirred vigorously at 45 °C for 3 h. Finally, the mixed solution was dialyzed and freeze-dried to obtain ALG-NHS powder.

For CNCs, 10 g α-cellulose was uniformly dispersed in 100 mL concentrated sulfuric acid solution (60 wt%) and stirred at 45 °C for 3 h under N$_2$ atmosphere. After adding cold water to terminate the reaction, the mixture was centrifuged, washed, dialyzed, and freeze-dried to obtain CNCs white powder.

**Preparation of Janus hydrogel.** First, a hydrogel precursor solution containing 4 mL deionized water, 0.6 g AM, 4.5 mg MBA, 0.03 g CNCs, 0.125 g Alg-NHS, 0.02 g Irgacure 2959, and a certain amount of SBMA was prepared. After degassing and deoxygenation, the precursor solution was transferred into a specially designed unilateral light-transmitting mold. Subsequently, 365 nm UV irradiation was vertically irradiated onto the custom-made mold for 60 min to obtain the Janus hydrogel. The weight ratios of SBMA and AM in the hydrogel were 1:1, 1:2, and 1:4, respectively. In addition, pure PAM and pure PSBMA hydrogels were obtained by the same procedure for comparison.

**Fabrication of Janus hydrogel bioelectronic interface.** A specially designed hollow-patterned waterproof mask was employed to customize Janus hydrogel-based personalized bioelectronic interfaces. Specifically, the waterproof mask-covered Janus hydrogel was immersed in a Py solution with varying concentrations (1, 2, and 3 wt%) at 4 °C. Subsequently, FeCl$_3$ (0.5 mmol·L$^{-1}$) was added to the solution and initiated Py polymerization at room temperature to construct an electronic conductive path.

**Structural, physicochemical properties and gelation mechanism characterization.** The structural aspects and physicochemical properties of the hydrogel were investigated with scanning electron microscopy (SEM, SU8010, HITACHI), Fourier transform infrared spectroscopy (FTIR, INVENIO-S, Bruker), X-ray photoelectron spectroscopy (XPS, Nexsa, Thermo Fisher), and two-photon laser confocal microscopy (LSM880, ZEISS). Complete, uncropped scanning electron microscopy (SEM) images have been deposited in the Source Data file to ensure transparency of sample characterization.

To elucidate the gelation mechanism, a micro infrared rheometer (MARS60, Thermo Scientific) equipped with a UV light curing module was employed to record real-time rheological properties and real-time FTIR spectroscopy during the gelation process.

### Performance characterization and evaluation

**Mechanical and rheological performance.** Uniaxial tension and compression properties of the hydrogel were measured with an electronic universal testing machine (WDW-01A, Kangyuan). Rheological properties of the hydrogel were evaluated with a rheometer (MARS60, Thermo Scientific) by frequency, temperature, and strain scanning.

**Adhesive performance.** For lap shear and 90° peel tests, an electronic universal testing machine (WDW-01A, Kangyuan) was used to record changes in shear/peel force and displacement in real time. Specifically, the Janus hydrogel and biological tissue were tightly bonded together. Rigid backings were adhered to the opposite sides of the hydrogel and biological tissue to prevent elastic deformation and mechanical fracture during testing. Subsequently, the electronic universal testing machine applied a uniaxial tension to separate the hydrogel from the biological tissue while simultaneously recording changes in shear/peel force and displacement.

For the burst pressure testing, a specially designed testing device was employed with fresh pigskin as the tissue model. Briefly, a circular defect ($\Phi = 5\,mm$) was created on the pigskin and secured onto a pressure chamber. Subsequently, Janus hydrogel was applied to fill and seal the defect. Finally, a gas pump was used to slowly pressurize the sealed pressure chamber, with real-time pressure changes monitored and recorded by a digital pressure gauge. The burst pressure was defined as the maximum pressure value recorded during the entire pressurization process.

### Electrical and electrochemical performance

**Conductivity.** 4-Point Probes Resistivity Measurement System (PROBES TECH, RTS-8) was employed to measure the resistance and calculate the conductivity ($\kappa$, S·m$^{-1}$).

**Electrochemical Performance.** All electrochemical tests were conducted via an electrochemical workstation (Zennium Pro, Zahner). Hydrogel interface, pure platinum (Pt) foil, and silver/silver chloride (Ag/AgCl) electrodes served as working electrodes, counter electrodes, and reference electrodes, respectively. PBS solution was utilized as the electrolyte solution.

For the CSC test, cyclic voltammetry was employed at a scan rate of 0.15 V·s$^{-1}$ and in a potential range of $-0.4$-$0.5$ V (vs. Ag/AgCl electrode). For the CIC test, the recurrent potential pulse method was employed at a biphasic cyclic pulse of $\pm0.5$ V (vs. Ag/AgCl). Notably, the timescale was consistent with ion diffusion over ~$10$–$20\,\mu m$ in hydrated hydrogel matrices, which matched the interfacial length scales relevant to charge-transfer at the hydrogel-electrode interface in vivo. For EIS, a sinusoidal AC voltage with 5 mV amplitude was applied in a frequency range from 10 kHz to 0.1 Hz.

### Application of Janus hydrogel bioelectronic interfaces

**In vitro study of cell viability and proliferation.** For cytotoxicity assay, mouse fibroblasts (L929) were employed to assess the biocompatibility of hydrogels by MTT assay, live/dead cell staining assay, and cell flow cytometry. For evaluation of cell proliferation and migration behavior accelerated by electrical stimulation, L929 cells were applied with a 100 mV/mm DC voltage at 25 Hz through a biological signal acquisition and processing system (MADLAB-4C/501H). Inverted fluorescence microscopy (ECLIPSE Ts2R, Nikon) was used to observe and record cell migration and proliferation behavior. Cell flow cytometry and cell counting kit-8 (CCK-8) assay were selected to quantitatively analyze cell proliferation and migration behavior.

**Western blotting.** Proteins were separated by 10% sodium dodecyl sulfate-polyacrylamide gel electrophoresis (SDS-PAGE) and transferred onto a polyvinylidene fluoride membrane. After washing with Tris-Borate-Sodium Tween-20 (TBST), the membrane was blocked with 5% non-fat milk in TBST. Subsequently, the membrane was incubated overnight at 4 °C with primary antibodies targeting EGF (1:1000, Beyotime Biotechnology), SMAD 2/3 (1:1000, Beyotime Biotechnology), or TGF-$\beta$ 1/2 (1:1000, Beyotime Biotechnology), respectively. Following incubation with secondary antibodies, the protein bands were visualized using an ultra-sensitive ECL chemiluminescence reagent kit (BeyoECL Plus, Beyotime Biotechnology). All complete scans of the key immunoblot membranes with clearly labeled positions of molecular weight/size markers have been provided in the Source Data file.

**Rat neuromodulation.** Sprague-Dawley (SD) rats (8 weeks, female) with an average weight of 200-250 g were used for in vivo neuromodulation experiments. Following anesthesia, the sciatic nerve was exposed via blunt dissection of the gluteal muscles. Janus hydrogel interface was then affixed to the sciatic nerve for neuromodulation. A biological signal acquisition and processing system (MADLAB-4C/501H) was used to deliver a series of monophasic rectangular pulses with varying voltages (0.15-1.0 V) and frequencies (5-50 Hz). Ankle joint angle changes under different stimulation intensities and frequencies were recorded, alongside electroneurogram (ENG) signals from the sciatic nerve in response to mechanical stimuli applied to the hind paw (strong beat, weak beat, and light tap).

**Abdominal wall defect model establishment and electrical stimulation therapy.** Twelve SD rats (8 weeks, female) with an average weight of 200-250 g were selected to establish full-thickness abdominal wall defect models. Briefly, SD rats were anesthetized with isoflurane and shaved abdominal hair. Subsequently, circular full-thickness abdominal wall defects ($\Phi = 10\,mm$) were surgically created under aseptic conditions using standard surgical procedures.

To verify the effect of electrotherapy, Janus hydrogel bioelectronic interface or commercially available polypropylene (PP) patch was attached to the abdominal wall defect. Three treatment groups were used to compare the effect of abdominal wall healing: (1) PP mesh coverage. (2) Hydrogel coverage. (3) Hydrogel coverage + electrical stimulation (ES). The electrical stimulation instrument and setting parameters for abdominal wall healing were consistent with the parameters of cell proliferation and migration induced by electrical stimulation.

### Ethics approval

This study was received and approved by the Special Committee on Research Ethics of Jiangsu Normal University (Approval No. JSNU-H2024CG-J011). All study subjects agreed to the use of their data for scientific research by signing the informed consent form and were informed of their right to withdraw their consent at any time. The authors affirm that human research participants provided written informed consent for publication of the images in Fig. 5g, h, and Supplementary Fig. S40.

All animal experiments were conducted with the assistance of Zhejiang Provincial People's Hospital and approved by the Animal Ethics Committee (Approval No. 20250505116584).

### Statistical analysis

All results of this study were derived from mean ± standard deviation (SD). Each experiment was performed at least 3 times with biological replicates. Comparisons between two groups were analyzed using two-tailed unpaired Student's $t$-test, following prior assessment of variance homogeneity via $F$-test: equal variances assumed, pooled variance $t$-test was applied; unequal variances, Welch's $t$-test was used. Differences were considered statistically significant at $P < 0.05$. All statistical analyzes were conducted using OriginPro Learning Edition 2024, GraphPad Prism 10.6, and ImageJ 1.53.

### Reporting summary

Further information on research design is available in the Nature Portfolio Reporting Summary linked to this article.

## Data availability

All data supporting the findings of this study are available within the article and its Supplementary files or from the corresponding author upon request. Source data are provided with this paper.

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

## Acknowledgements

The work was supported by the National Natural Science Foundation of China (No. 62288102 to X.D.) and Key Project of Basic Research Program of Jiangsu Province (BK2024303 to X.D.).

## Author contributions

X.D., Z.N., Q.W. and X.Q. designed the project and performed the experiments. X.Q., H.S and D.G. prepared and characterized the materials. X.Q. wrote the manuscript with support from X. D., Z.N. and Q.W. Y. Z. performed molecular dynamics simulation. All authors contributed to the data analysis, discussed the results, and commented on the manuscript.

## Competing interests

The authors declare no competing interests.
