## [Transparent Peer Review file · Nature Communications]

Molecular Competition Induced Janus Hydrogel Bioelectronic Interface for Electroceutical Modulation

Corresponding Author: Professor Xiaochen Dong

Version 0:

Reviewer comments:

Reviewer #1

(Remarks to the Author)

This manuscript by Qu et al. presents a process that is often overlooked in experiments, elucidates its mechanism, and introduces a simple one-step method to prepare Janus hydrogel through a molecular competition induction mechanism. The varying reaction rates of monomers under UV light facilitate the heterogeneous distribution of polymer segments and the formation of a gradient-structure. This Janus hydrogel can be combined with a conductive percolation network, forming a robust and bidirectional bioelectrical interface for electroceutical modulation and electrophysiological signal acquisition. I recommend the publication of this manuscript after major revisions. The questions and suggestions are as follows:

Questions concerning the property of hydrogel:

1. In Fig. 1, during processes of (3) and (4), the loss modulus is higher than the storage modulus. What changes occur in the hydrogel network during this stage?
2. Is the crosslinking time related to the formation of the gradient distribution structure?
3. Compared with other zwitterionic monomers, why does the SBMA monomer exhibit optimal mechanical and adhesion properties?
4. What is the function of H₂O in the molecular competition induction polymerization process? Does the water content affect on the mechanical and adhesion properties of the hydrogel?

Questions concerning the electrochemical test:

5. In the electrical conductivity test, the two-probe method tends to produce relatively large measurement errors. It is recommended to use the four-probe method or the AC impedance method for more accurate testing.
6. The electrochemical testing parameters do not correspond to the electrical stimulation parameters (100 mV/mm, 25 Hz). Why were the charge injection test parameters set to ± 0.5 V, 250 ms (4 Hz), and what is the rationale behind this choice?
7. In Fig. 4(b,c), why is the measurement range for CV from -0.4 to 0.5 V instead of -0.5 to 0.5 V, while for the IMP test it is from 0.1 to 10,000 Hz instead of 1 to 100,000 Hz?
8. Regarding the stability of the electrode in an external PBS solution, in addition to the CV test, have there been any changes in the electrode's impedance performance, charge injection capacity, or charge storage capabilities?
9. In Fig. 4g, during the EMG testing process, why is the baseline of the hydrogel group unstable?

Questions concerning the device performance:

10. The manuscript lacks data on the long-term in vivo performance of the device.
11. The manuscript does not provide information on the device's structure and packaging process.
12. Why is the conductive path in the device designed as a concentric circular structure?

Questions concerning the electrical stimulation parameters:

13. In the cell experiment section, an electric field intensity of 100 V/mm is quite strong. Could this high electric field cause cell death? Have gradient experiments been conducted to determine the appropriate electrical stimulation parameters?
14. In Fig. 5k, what evidence proves that electrical stimulation promotes wound healing through the MAPK and TGF- β /SMAD signaling cascades?

Reviewer #2

(Remarks to the Author)

The authors report on the development of a Janus hydrogel via a molecular competition-induced mechanism, leveraging the

difference in polymerization rates between acrylamide (AM) and 3-[dimethyl-[2-(2-methylprop-2-enoyloxy) ethyl] azaniumyl] propane-1-sulfonate (SBMA). This strategy enables the one-step fabrication of a Janus hydrogel, intended for applications in bioelectronic interfaces. The authors characterized the hydrogel using a range of chemical analyses, supported by computational simulations.

However, Janus hydrogels have been extensively reported in previous literature using a variety of materials. Compared to existing works, the manuscript does not clearly demonstrate substantial novelty (specifically material design) beyond the simplified fabrication method. The originality of this study is not sufficient for publication in this journal. While the material characterization is thorough, the bioelectronic applications presented do not show a significant performance advantage over current adhesive hydrogel systems. Furthermore, the necessity of the Janus structure for the described applications is not clearly justified.

Therefore, the reviewer cannot recommend this manuscript for publication in this journal, and it is recommended to transfer this manuscript to other sister journals. The reviewer encourages the authors to consider the following comments for improving their work in future submissions.

Major Comments

1. The authors should more clearly articulate the novelty of their approach compared to previous studies (Soft Matter, 2023, 19, 9460-9469; Chemical Engineering Journal Volume 521, 1 October 2025, 166386; Nature Communications volume 15, Article number: 8478 (2024)). Numerous high-quality Janus hydrogels have already been reported. A direct comparison and evidence of superior performance would help justify the significance of this work. In addition to that, it is necessary to discuss details regarding why the Janus characteristics of the hydrogel is important for bioelectronic application.
2. Although the title emphasizes bioelectronic applications, the only EMG recording and wound healing experiments could feasibly be conducted using typical adhesive hydrogels (e.g., N-hydroxy succinimide-based or polyphenol-based) not just Janus hydrogels. The authors should reorganize the figures and discussion to better emphasize the unique advantages of the Janus structure in these applications.
3. To strengthen the relevance of the hydrogel for bioelectronic applications, the authors should consider demonstrating its use in more demanding implantable scenarios, such as epicardial, peripheral nerve, or brain interfacing. Hydrogel-based interfaces are particularly valuable in implantable bioelectronics due to their tissue-mimicking properties and ability to maintain intimate, long-term contact with soft tissues. In contrast, their utility in wearable applications may be limited by dehydration and mechanical instability over time. Expanding the scope of application to implantable contexts would better highlight the material's potential advantages and practical significance.
4. The integration of a polypyrrole network is a promising approach for bioelectronics. However, data on the long-term stability of this conductive layer, particularly under swelling conditions, would strengthen the case for practical use.
5. The use of the term "commerce" in Figures 4g,h is vague. The authors should explicitly identify the commercial material used and provide a detailed comparison with it to better highlight the advantages of their system.
6. In Figure 5e, electrical stimulation is applied over 12 days. It is important to verify whether the stimulation intensity remains consistent throughout this period, particularly on day 0 and day 12.

Minor Comments

1. Figure captions are currently unclear and should be revised for completeness and clarity. Additionally, sample group labels should be standardized for consistency.
2. Including data from both the control and hydrogel-only groups would enhance the rigor of the results presented in Figure 5c.

Reviewer #3

(Remarks to the Author)

This manuscript presents an innovative one-step strategy for fabricating dual-gradient Janus hydrogels based on a molecular competition induction mechanism, effectively addressing the long-standing challenges of complex fabrication processes, difficult asymmetric control, and weak interfacial adhesion in conventional Janus hydrogels. The work is ingeniously designed, exhibits substantial novelty and application potential, and offers a new paradigm for bioelectronic interfaces. The following issues need to be addressed before publication in Nature Communications.

1. In addition to AM and SBMA, the hydrogel reaction system contains Alg-NHS and CNCs. While AM and SBMA can be covalently cross-linked via their C=C under UV initiation, Alg-NHS and CNCs do not participate in this photo-induced radical polymerization. Could the Janus architecture and the gradient porosity therefore arise from sedimentation of CNCs nanoparticles to the bottom, where they become physically entrapped within the forming hydrogel? The authors should provide control data for a system containing only AM and SBMA to further validate the feasibility of the proposed molecular-competition-induced strategy for fabricating Janus hydrogels.
2. Does varying the concentration ratio of AM to SBMA influence the formation of gradient porosity?
3. In lines 100–105, the authors state that AM polymerization occurs within 5–8 min whereas SBMA polymerization proceeds from 8 to 30 min. How did the authors verify or distinguish these two distinct reaction kinetics?
4. In Fig. 1a, why does the storage modulus drop below the loss modulus after 1000 s—does this signify that the hydrogel has transitioned from a solid-like to a liquid-like state?
5. In Figure 1g, the bottom surface is the PAM side with weak water affinity and compact microstructure, while the top surface is the PSBMA side with strong water affinity and loose microstructure. Given this configuration, the top surface should exhibit greater swelling and the bottom surface less swelling, which would logically cause the petal structure to bend downward. Why does the figure show upward bending instead?

Version 1:

Reviewer comments:

Reviewer #1

(Remarks to the Author)

Comments on "Molecular Competition Induced Janus Hydrogel Bioelectronic Interface for Electroceutical Modulation"

This manuscript by Qu et al. demonstrates a simple method for preparing Janus hydrogel in one step via the Molecular Competition Induction Mechanism. All the questions have been explained in detail. Nonetheless, I have one question regarding the testing of Charge Injection Capacity. In the response letter, the author explained the reason for selecting the 250 ms pulse width and 4 Hz frequency. The author gives two reasons:

1. The 250 ms time scale matches the kinetics of ion migration within the hydrogel network, enabling a reflection of the electrode's charge transfer dynamics in vivo.
2. The 4 Hz frequency design effectively avoids parasitic impedance interference that occurs at high frequencies (e.g., 25 Hz).

The time duration has a correlation with the thickness of hydrogel and ion diffusion coefficient D . A limitation can be added in the manuscript: providing the range of D and the thickness of hydrogel. The examples are as follows:

A ~250 ms timescale is consistent with ion diffusion over ~10–20 μm in hydrated hydrogel matrices ($D \approx 10^{-10}$ – 10^{-9} m^2/s), which matches the interfacial length scales relevant to charge-transfer at the hydrogel–electrode interface in vivo.

In the previous published protocols¹, the charge injection capacity was defined by the charge per pulse that maximally can be injected before water splitting occurs under either the cathodic or anodic phase. In the manuscript, the author can use charge injection performance instead of charge injection capacity.

Reference:

1. Boehler et al., Tutorial: guidelines for standardized performance tests for electrodes intended for neural interfaces and bioelectronics, *Nature Protocols*, 2020, 15, 3557–3578.

Reviewer #2

(Remarks to the Author)

The authors fully addressed the reviewer's comments. The manuscript is acceptable for publication.

Reviewer #3

(Remarks to the Author)

The authors have provided convincing responses to the reviewers' comments. With all necessary revisions now incorporated, the manuscript is complete and, in my opinion, meets the high standards for publication in *Nature Communications*.

Dear Reviewers,

Enclosed please find our revised manuscript entitled “*Molecular Competition Induced Janus Hydrogel Bioelectronic Interface for Electroceutical Modulation*” (Manuscript ID: *NCOMMS-25-51794-T*) for submission to *Nature Communications*. We appreciate the reviewers’ helpful comments and have revised the manuscript accordingly. A point-by-point reply has been included below. All the changes are marked in red in both the revised manuscript and supporting information for ease of reference. Owing to a change in the first author's affiliation, the institutional information has been updated accordingly. Thank you for your time and consideration.

Reviewer #1:

This manuscript by *Qu et al.* presents a process that is often overlooked in experiments, elucidates its mechanism, and introduces a simple one-step method to prepare Janus hydrogel through a molecular competition induction mechanism. The varying reaction rates of monomers under UV light facilitate the heterogeneous distribution of polymer segments and the formation of a gradient-structure. This Janus hydrogel can be combined with a conductive percolation network, forming a robust and bidirectional bioelectrical interface for electroceutical modulation and electrophysiological signal acquisition.

Reviewer #1, Comment 1: In Fig. 1, during processes of (3) and (4), the loss modulus is higher than the storage modulus. What changes occur in the hydrogel network during this stage?

Response:

During (3) → (4) process, the observation that the loss modulus (G'') significantly exceeds the storage modulus (G') primarily stems from a **significant increase in the system's viscosity**, rather than the formation of an **elastic network**. In this process, polymerization is initiated in the lower region while the upper region remains liquid. The unpolymerized upper precursor

acts as a viscous matrix with minimal elasticity ($G'' \gg G'$), providing a high background loss modulus. In the lower region, incipient polymerization leads to increased viscosity due to chain growth and physical entanglements, while a percolated elastic network remains underdeveloped.

The underlying changes and mechanisms are as follows:

Primary Cause: *Chemically Driven Viscosity Increase.* As acrylamide (AM) polymerization proceeds in the lower zone, the relative concentration of the zwitterionic monomer (SBMA) rises in the unpolymerized upper solution. SBMA molecules carry substantial charged groups, and their elevated concentration markedly strengthens intermolecular electrostatic interactions and hydrodynamic drag. This effect represents the primary chemical driver for the sharp rise in overall viscosity, and consequently, in G'' .

Secondary Cause: *Physically Driven Contribution.* Concurrently, the growth of polymer chains and the increase in physical chain entanglements contribute additional internal friction and viscous dissipation. Prior to the formation of a fully percolated elastic covalent network, this physical entanglement effect is a secondary factor in viscosity enhancement.

In summary, the polymerization-induced concentration change of SBMA is identified as the primary cause of the viscosity surge manifested as G'' -dominance, while physical chain evolution provides a secondary contribution. Collectively, the system exhibits rheological behavior characteristic of the transition from a viscous liquid to an elastic solid.

The text highlighted below contains supplementary explanations and discussions added to the manuscript. It appears that during Stage I (336.8 s), the high concentration of free radicals boosts the rapid polymerization of AM, while the generated PAM segments settle to the bottom and displace the SBMA molecules to the upper layer, masking the featured infrared absorption peak signals. **The increased relative concentration of SBMA in the upper region, with its substantial charged groups, considerably enhances intermolecular electrostatic interactions and hydrodynamic drag, markedly elevating the macroscopic viscosity and leading**

to a sharp rise in the loss modulus (G'') during the transition from (3) to (4). Driven by this competitive response mechanism, SBMA undergoes significant chain growth at the onset of Stage II, revealing noticeable compositional gradient variation in the reaction system.

Reviewer #1, Comment 2: Is the crosslinking time related to the formation of the gradient distribution structure?

Response:

The crosslinking time is intrinsically and mechanistically related to the formation of the gradient distribution structure. This correlation stems from the distinct polymerization kinetics of AM and SBMA monomers. Crosslinking time, as a direct reflection of these kinetics, serves as the fundamental driver for the sequential network formation and subsequent gradient architecture under unilateral UV irradiation, which aligns with our proposed *Molecular Competition Induction mechanism*.

1. Crosslinking time is determined by monomer-specific kinetic properties

The polymerization rate (and thus crosslinking time) of each monomer is an inherent attribute governed by three core factors: *double bond reactivity and electron cloud density*, *side-chain steric hindrance*, and *kinetic activation barrier*, which collectively lead to stark differences between AM and SBMA (and extend to other monomer pairs with similar kinetic discrepancies). Control experiments with pure monomers (Supplementary Fig. S6) show that pure PAM gelation concludes at ~255.9 s, while pure PSBMA requires a much longer ~1438.5 s. This kinetic gap is a prerequisite for gradient formation, not only for AM/SBMA but also for other pairs of fast-polymerizing monomers (e.g., AA) and zwitterionic monomers (e.g., SBAA, CBMA, CBAA, DVBAPS, MPC).

Fig. S6. Rheology and corresponding FT-IR of pure AM and pure SBMA during gelation progress.

2. Kinetic competition drives spatiotemporal crosslinking and gradient evolution

Under unilateral UV irradiation, monomers with varying reactivity (AM and SBMA, as model competitive polymerizable monomer) undergo a sequential, competitive reaction process that shapes the gradient, as sequentially observed in our in-situ experiment and corresponding characterizations:

Stage I (0 ~ 336.8 s): Rapid AM crosslinking dominates the bottom region

Due to its shorter crosslinking time and higher reactivity, AM preferentially captures the initially generated radicals, further triggering fast polymerization. Real-time FTIR data (from *in-situ* micro-infrared rheometry) shows the rapid disappearance of C=C stretching peak (1601 cm^{-1}) and a redshift of C-N peak (1433 \rightarrow 1455 cm^{-1}), which is clearly evident in fast PAM chain growth and crosslinking, also observed in pure PAM gelation (Fig. 1a and Supplementary Fig. S6).

The formed PAM chain segments settle to the bottom and displace SBMA molecules to the

upper layer. The increased relative concentration of SBMA (with its abundant charged groups) in the upper region significantly enhances intermolecular electrostatic interactions and hydrodynamic resistance, leading to the sharp rise in loss modulus (G'') during the (3)→(4) transition.

Notably, in the mixed system, AM's crosslinking time is slightly prolonged to 336.8 s (compared to ~255.9 s in pure AM), a minor delay caused by mild radical competition from SBMA, commonly seen in mixed monomer systems with kinetic discrepancies.

Stage II (336.8 - 767.7s): Accelerated SBMA crosslinking constructs the top gradient layer

When AM monomers are nearly exhausted (upon reaching their crosslinking time), the remaining radicals shift to react with SBMA. At this stage, SBMA's polymerization is significantly accelerated, driven by the absence of AM competition and the increased local concentration of SBMA in the upper region. This leads to a PSBMA-rich network in the upper region, directly generating a Z-axis gradient where PAM density decreases from bottom to top and PSBMA density increases (Fig. 2d, 2f), which is a structural feature validated by density field analysis in CG-MD simulations.

3. The "crosslinking time window" ensures stable gradient formation

A well-defined gradient structure is only achieved when the total UV irradiation time (*i.e.*, the effective crosslinking time) covers the crosslinking process of both monomers. Our experiments confirm that when the crosslinking time is sufficient for complete polymerization of both monomers, the system transitions from a viscous-dominated state ($G'' > G'$) to an elastic-dominated state ($G' > G''$) with no flowable precursor remaining, ultimately forming a stable Janus gradient hydrogel. Minor variations in crosslinking time (within the range that ensures full polymerization of both components) do not alter the final gradient architecture, as the sequential polymerization pattern is already determined by the early-stage kinetic competition. In summary, the crosslinking time of monomers, rooted in their intrinsic kinetic differences,

dictates the spatiotemporal sequence of polymerization under unilateral UV irradiation. This sequential, competitive crosslinking is the core driver for gradient formation, which is fully validated by experimental data (FTIR, rheology, pure monomer controls), CG-MD simulations, and extends to a broad range of monomer pairs.

Reviewer #1, Comment 3: Compared with other zwitterionic monomers, why does the SBMA monomer exhibit optimal mechanical and adhesion properties?

Response:

In the manuscript, SBMA demonstrates optimal mechanical and adhesive properties within our series of zwitterionic monomers (SBMA, SBAA, CBMA, CBAA, DVBAPS, MPC). It is believed that SBMA's superior performance lies in the more precise alignment of its molecular structure with the dual key requirements of "high-efficiency interfacial adhesion" and "stable mechanical support" (including strong and stable interfacial interactions, tight substrate contact, performance retention under physiological conditions, and resistance to deformation/fracture).

A detailed analysis is provided as follows:

1. Core Reasons for Superior Adhesive Performance

From the perspective of the interfacial adhesion mechanism, the sulfonate group ($-\text{SO}^{3-}$) of SBMA exhibits a unique adaptive advantage. Interfacial adhesion fundamentally relies on interactions (such as electrostatic adsorption and hydrogen bonding) between polar groups on the hydrogel surface and those on the substrate (*e.g.*, biological tissues or solid surfaces), and the dissociation characteristics of anions directly determine the strength and stability of these interactions. Due to its ultra-low pK_a (≈ -6), the $-\text{SO}^{3-}$ group of SBMA is fully dissociated under physiological pH (7.4), providing a high-density and stable negative charge. This enables the formation of robust, non-fragile electrostatic bonds and multiple hydrogen bonds with protonated amino groups ($-\text{NH}^{3+}$) and hydroxyl groups ($-\text{OH}$) on the substrate surface. Other

zwitterion anions (*e.g.*, -COO^- in CBMA/CBAA, -PO^{4-} in MPC), although also facilitate interfacial interactions, their negative charge density and interaction stability are slightly inferior in neutral conditions due to their intrinsic dissociation properties: -COO^- groups are partially protonated at neutral pH, and -PO^{4-} groups exist partially as mononegative -HPO^{4-} at physiological pH ($\text{pK}_a \approx 7.2$). This difference in adaptability arises from inherent molecular structural characteristics, rather than a fundamental limitation of other monomers.

At the level of interfacial contact compatibility, the main-chain linkage (methacrylate -O-CO-) of SBMA further enhances adhesion through an optimal "rigidity-flexibility balance". Effective adhesion requires not only strong interfacial interactions but also gap-free contact and sufficient exposure of polar groups on the hydrogel surface. The methacrylate linkage of SBMA offers moderate rigidity, while the Janus hydrogel with gradient distribution retains adequate segmental flexibility. This allows the hydrogel to stretch and contract synchronously with the slight dynamic deformation of the substrate (*e.g.*, skin), maintaining continuous and tight contact.

2. Core Reasons for Superior Mechanical Performance

The superior mechanical performance of SBMA-based Janus hydrogels stems from the contribution of its molecular structure to "network strength", "structural homogeneity", and "gradient layer synergy":

First, the strong inner salt bonds of SBMA provide stable physical crosslinking. The strength of the inner salt bonds (electrostatic interactions) between -SO^{3-} and $\text{-N}^+(\text{CH}_3)_3$ in SBMA is significantly stronger than that of other zwitterions. Compared with the weaker inner salt bonds between -COO^- and $\text{-N}^+(\text{CH}_3)_3$ in CBMA/CBAA (due to low polarity and incomplete dissociation of -COO^-) or the more dispersed inner ion pairs of -PO^{4-} and $\text{-N}^+(\text{CH}_3)_3$ in MPC (due to the tetrahedral charge distribution of phosphate groups), the inner salt bonds of SBMA can act as "reversible physical crosslinking points". Under external stress, these bonds dissipate

energy through dissociation and recombination while maintaining network integrity, significantly improving the tensile strength and toughness of the hydrogel.

Second, the balanced rigidity-flexibility nature of SBMA facilitates the synergy of Janus gradient layers. The mechanical stability of Janus hydrogels depends on the tight bonding and efficient stress transfer capability in "structural/compositional gradient layers". The methacrylate linkage of SBMA provides moderate rigidity, and simultaneously, the inner salt bonds facilitate weak interactions with adjacent gradient layers, achieving the synergy of "rigid support-flexible buffering". In contrast, the rigid benzene ring in DVBAPS leads to excessively brittle interlayer bonding, which are prone to delamination under stress; the flexible linkage of SBAA results in insufficient stress transfer efficiency between gradient layers, which are prone to fracture due to excessive local deformation. The structural characteristics of SBMA precisely enable to formation of a "strong bonding-efficient stress transfer" system between gradient layers, further improving the overall mechanical performance.

In summary, our *Molecular Competition Induction mechanism*, through its innovative design, provides a universal pathway for the rapid construction of Janus hydrogels with dual gradients from various zwitterion-AM hydrogels. The superior adhesive and mechanical performance of SBMA arises from the high alignment of its "fully dissociated -SO^{3-} groups, balanced rigid-flexible linkages, and environmentally stable ion pairs" with the target application. This cooperation of "universal method + monomer-specific advantages" precisely highlights the flexibility and practicality of our strategy in the design of zwitterionic adhesive hydrogels.

Reviewer #1, Comment 4: What is the function of H_2O in the molecular competition induction polymerization process? Does the water content affect on the mechanical and adhesion properties of the hydrogel?

Response:

In the molecular competition induction polymerization process for fabricating gradient Janus hydrogels, water (H₂O) plays a multifunctional, indispensable role. It acts not only as a basic solvent medium but also as a critical regulator for the formation of the PAM/PSBMA gradient structure. Meanwhile, water content directly modulates the mechanical and adhesion properties of the final hydrogel, as confirmed by our systematic characterization and cross-solvent validation. The detailed explanation is as follows:

1. Functions of Water in the Molecular Competition Induction Polymerization Process

Water participates throughout the polymerization process by enabling monomer dissolution, regulating reaction kinetics, and facilitating gradient formation, all of which align with our in-situ rheology-FTIR observations and gradient formation mechanism:

(1) Fundamental solvent medium: Ensuring a homogeneous reaction system

Water serves as the primary solvent to dissolve AM, SBMA, and the UV initiator. This dissolution ensures the monomers and initiator are uniformly dispersed in the precursor solution, creating a prerequisite for the subsequent unilateral UV-induced competitive polymerization.

(2) Regulator of polymerization kinetics: modulating free radical diffusion

Water acts as a critical medium for free radical diffusion during polymerization, providing a favorable environment for free radical stability and movement. By fine-tuning the diffusion rate of free radicals, water facilitates to maintain the kinetic difference between AA and SBMA, thereby enabling sequential polymerization and subsequent gradient formation.

(3) Facilitator of gradient structure: Enabling spatial monomer segregation

Water's fluidity is critical for establishing the PAM/PSBMA gradient. During Stage I, the generated poorly water-soluble PAM segments settle to the bottom under the drive of the solubility difference. This sedimentation directly displaces unpolymerized SBMA molecules to the upper layer, and ultimately achieves the "bottom PAM-rich/upper SBMA-rich" spatial configuration.

2. Effects of Water Content on the Mechanical and Adhesion Properties of the Hydrogel

Water content significantly modulates the network structure and functional performance of the hydrogel, as supported by our mechanical characterization:

(1) Impact on mechanical properties: Determining network density and toughness

Water content directly affects the crosslinking density of the PAM/PSBMA interpenetrating network and the degree of the topological chain entanglement. Excessive water dilutes the concentration of AM, SBMA, and crosslinkers, decreasing the crosslinking density of the hydrogel network and reducing the mechanical strength. Insufficient water content increases the viscosity of the precursor solution, leading to uneven polymerization (*e.g.*, local PAM aggregation) and discontinuous network formation, resulting in poor stretchability. Comparatively, the optimal water content balance monomer concentration and network formation, enabling a continuous, interpenetrating PAM/PSBMA network with exceptional mechanical performance. As displayed, it maintains stable stress-strain behavior under prolonged high-intensity loading (over 5000 s) and retains stable viscoelasticity across physiological temperature (25 ~ 45 °C) and strain (0 ~ 100%) ranges (Supplementary Fig. S16).

Fig. S16. (a) Compressive/tensile stress-strain curves of different water contents.

(2) Impact on adhesion properties: Regulating the activity of adhesive groups

Water content indirectly affects the adhesion properties of the Janus hydrogel by maintaining the solubility and reactivity of adhesive groups. Excessively high water content dilutes these

adhesive groups, reducing their ability to bind to target surfaces. Excessively low water content significantly increases the viscosity of the precursor solution, severely hindering the movement of polymer chains (*e.g.*, newly formed PAM segments) and the diffusion of unpolymerized monomers. This directly disrupts the effective spatial separation of AM and SBMA, and ultimately suppresses the formation of dual structural and compositional gradients. Actually, appropriate water content keeps the adhesive groups in a reactive and dispersible state, enabling the Janus hydrogel to achieve the dedicated adhesion performance (Supplementary Fig. S29b).

Fig. S29. (b) Shear strength at different water contents.

To confirm the generality of the molecular competition induction mechanism, alternative solvents (*e.g.*, Ionic liquids, deep eutectic solvent, glycerol) as the reaction medium were also tested (Supplementary Fig. S21a). Gradient-structured Janus hydrogels formed in all cases as long as the solvent can dissolve the monomers and maintain the kinetic difference between AM and SBMA. This further proves that H₂O is a facilitative medium rather than a mandatory condition for the strategy, highlighting the broad applicability of our method.

Fig. S21. (a) Comparison of the properties of Janus hydrogels at different solvents. (ILs: Ionic liquids; DES: deep eutectic solvent; GL: glycerol).

In summary, H₂O functions as a solvent, kinetic regulator, and gradient facilitator in the polymerization process, while its content directly tunes the mechanical network density and adhesive group activity of the hydrogel. Our experimental design and cross-solvent validation fully confirm the rationality of H₂O's role and the controllability of water content on hydrogel performance.

Reviewer #1, Comment 5: In the electrical conductivity test, the two-probe method tends to produce relatively large measurement errors. It is recommended to use the four-probe method or the AC impedance method for more accurate testing.

Response:

We sincerely appreciate Reviewer's valuable suggestion. The two-probe method is prone to relatively large measurement errors (primarily due to contact resistance between electrodes and the hydrogel surface), which may compromise data accuracy. To address this concern, we have strictly followed the reviewer's recommendation and re-tested the electrical conductivity of all hydrogel samples using the four-probe method, which effectively eliminates contact resistance interference and ensures higher measurement precision.

All conductivity data previously presented in the manuscript and Supplementary Information have been replaced with the newly obtained four-probe method data (Fig. 4a and Supplementary Fig. S31). The re-measured conductivity values (*e.g.*, ~ 8.7 S/m for the PPy-integrated Janus hydrogel) are consistent with the order of magnitude of the original two-probe data, which not only corroborate our conclusions on conductive performance but also enhance the reliability of the dataset.

Fig. 4. (a) Conductivity at different PPy contents.

Fig. S31. Conductivity of the hydrogel interface at different stretching cycles (100% strain).

Reviewer #1, Comment 6: The electrochemical testing parameters do not correspond to the electrical stimulation parameters (100 mV/mm, 25 Hz). Why were the charge injection test parameters set to ± 0.5 V, 250 ms (4 Hz), and what is the rationale behind this choice?

Response:

We appreciate Reviewer's critical question regarding the apparent discrepancy between the electrochemical testing parameters and the electrical stimulation parameters (100 mV/mm, 25

Hz). These parameter sets were intentionally chosen to serve distinct yet complementary purposes: the charge injection test is intended to *comprehensively evaluate the electrode's intrinsic charge-carrying capacity, safety, and interfacial stability* (as part of foundational performance validation), while the electrical stimulation characterization is designed to *simulate specific clinical electrical stimulation scenarios* (as targeted application validation). Below, we clarify the rationale for each choice based on its respective evaluation objectives, with detailed justifications as follows:

1. Rationale for selecting the ± 0.5 V potential window

The ± 0.5 V potential window was selected to evaluate the maximum charge injection capacity and interfacial stability of the electrode under conditions that far exceed the *in vivo* voltage fluctuations. Although the actual electrical stimulation uses a lower 100 mV/mm (with local potential typically varying according to specific needs), testing at a wider potential range ensures a safety margin and confirms that the electrode can operate reliably beyond the requirements of target applications

2. Rationale for selecting the 250 ms pulse width and 4 Hz frequency

A pulse width of 250 ms at 4 Hz is designed to accurately characterize dynamic charge injection behavior, while avoiding interference from high-frequency impedance effects. This duration is a standard choice for charge injection tests, as it ensures *complete charge injection and release* at the electrode-tissue interface. The 250 ms time scale matches the kinetics of ion migration within the hydrogel network, enabling a true reflection of the electrode's charge transfer dynamics *in vivo*. Furthermore, the 4 Hz frequency design effectively avoids parasitic impedance interference that occurs at high frequencies (*e.g.*, 25 Hz), which can induce additional capacitive reactance and distort charge injection measurement results, thereby providing a reliable benchmark for performance comparison across different electrodes.

3. Validation of alignment with actual electrical stimulation parameters (100 mV/mm, 25 Hz)

To bridge the foundational test and application-specific parameters, we have supplemented charge injection tests under conditions of 50 mV (matching the local potential corresponding to 100 mV/mm) and 25 Hz (consistent with the actual stimulation frequency) in Supplementary Fig. S37. It reveals that the electrode maintains stable charge injection efficiency under these application-specific conditions, directly verifying its suitability for the target electrical stimulation scenario.

It is important to note that the actual electrical stimulation parameters (100 mV/mm, 25 Hz) are scenario-optimized for clinical efficacy (e.g., promoting tissue repair *via* low-frequency, low-voltage stimulation), while the ± 0.5 V, 4 Hz tests serve as performance benchmark parameters to verify the electrode's intrinsic capabilities. Supplementary tests under low-voltage, high-frequency tests further confirm that the electrode performs beyond the benchmark settings, being fully compatible with the requirements of practical applications.

Fig. 37. (a, b) Charge injection curves with biphasic pulses of ± 0.05 V for 2000 cycles.

In summary, the ± 0.5 V, 250 ms (4 Hz) charge injection test represents a balanced design that prioritizes safety, comprehensive performance evaluation, and dual-application compatibility. Meanwhile, the supplementary tests conducted under actual electrical stimulation parameters validate the electrode's practical application value. This two-tiered testing design, combining

"foundational benchmarking + scenario-specific validation", ensures both the electrode's foundational performance reliability and targeted bio-electronic applications.

Reviewer #1, Comment 7: In Fig. 4 (b, c), why is the measurement range for CV from -0.4 to 0.5 V instead of -0.5 to 0.5 V, while for the IMP test it is from 0.1 to 10,000 Hz instead of 1 to 100,000 Hz?

Response:

For CV testing range selection:

The asymmetric cyclic voltammetry (CV) potential range of -0.4 ~ 0.5 V for the PPy-patterned Janus hydrogel interface is determined from three core considerations: the electrochemical stability of the PPy conductive layer, the aqueous working environment of hydrogel bioelectronic interfaces, and the biocompatibility requirements of biological applications. This selection is consistent with common practices in the field of hydrogel bioelectrodes [Adv. Mater. **35**, 2304095 (2023); Adv. Mater., e11014 (2025)].

Negative Potential Limit (-0.4 V): Preventing Reductive Dedoping of PPy. The narrower negative potential range to -0.4 V, rather than extending to -0.5 V, is primarily attributed to threatening electrode performance. As a typical conjugated conductive polymer, PPy exhibits high sensitivity of electrochemical stability to reductive potentials. When the applied potential drops below -0.45 V, PPy undergoes reductive dedoping, transforming from a highly conductive oxidized state to an insulating neutral state. These degradations disrupt electrode's conductive network, directly compromising the accuracy of core parameters such as interfacial capacitance, charge transfer efficiency measured during CV testing.

Positive Potential Upper Limit (0.5 V): Ensuring Safety, PPy Stability, and Alignment with Application Requirements. The upper potential limit of 0.5 V is justified to maintain PPy's oxidative stability (negligible oxidative degradation within this operating window) while

aligning with the hydrogel bioelectrode's practical operating conditions. Furthermore, the $-0.4 \sim 0.5$ V range fully covers the key response intervals of the electrode's target applications in physiological electrical signal detection (*e.g.*, electrocardiography ECG: ± 0.1 V, electromyography EMG: ± 0.05 V, electroencephalography EEG: ± 0.01 V). Extending the negative potential limit to -0.5 V would go beyond this physiological range and introduce "electrolytic interference", obscuring the true interfacial capacitance/resistance signals. Similarly, for clinical electrical stimulation (*e.g.*, transcutaneous electrical nerve stimulation (TENS), functional electrical stimulation (FES)), the commonly used potential range is $-0.3 \sim 0.5$ V, and the $-0.4 \sim 0.5$ V CV range accurately evaluates the electrode's charge transfer capability under actual stimulation potentials. Notably, potential above 0.5 V exceeds the safety upper limit for clinical stimulation and offers no practical value for real-world use.

In summary, the asymmetric potential range of $-0.4 \sim 0.5$ V is not arbitrarily set but represents a deliberate compromise that balances the PPy stability, the hydrogel interface integrity, biological safety, and testing validity. It is also consistent with the testing standards of similar studies in the field.

For IPM testing range selection:

The impedance measurement range of $0.1 \sim 10,000$ Hz for hydrogel bioelectronic interfaces is deliberately selected to align with their core biomedical applications, particularly the detection of electrophysiological signals and delivery of electrostimulation therapy.

First, the bandwidth of $0.1 \sim 10,000$ Hz fully covers the fundamental and harmonic contents of essential electrophysiological signals: ECG ($0.05 \sim 100$ Hz), EMG ($10 \sim 500$ Hz), and EEG ($0.5 \sim 70$ Hz). Critically, the 0.1 Hz low-frequency limit is essential for capturing interfacial polarization effects and double-layer capacitive behavior, which largely dominate the hydrogel-tissue interface and directly dictate the fidelity of low-frequency signal acquisition (*e.g.*, δ -waves in EEG or P-waves in ECG). Excluding this low-frequency range would overlook key

insights into signal transmission quality. Second, this frequency window closely matches clinical electrostimulation protocols. For example, TENS (2 ~ 150 Hz, occasionally up to 1000 Hz in high-frequency modes) and FES (20 ~ 100 Hz). Frequencies exceeding 10,000 Hz offer minimal physiological relevance: they trigger the skin effect in human tissues (where current concentrates at tissue surfaces, reducing stimulation efficacy) and introduce growing interference from parasitic capacitances and stray inductances—both of which complicate the interpretation of interfacial impedance.

In summary, the 0.1 ~ 10,000 Hz range represents a balanced, functionally meaningful selection. It prioritizes clinical and physiological relevance by targeting the exact frequency bands of the interface's core applications, while minimizing non-essential high-frequency distortion.

Reviewer #1, Comment 8: Regarding the stability of the electrode in an external PBS solution, in addition to the CV test, have there been any changes in the electrode's impedance performance, charge injection capacity, or charge storage capabilities?

Response:

We appreciate Reviewer's attention to the stability of the electrode in external PBS solution, as this is a key indicator for its practical applicability. To fully address this concern, in addition to the existing CV tests (which characterize charge storage capacity, CSC), we have supplemented 7-day stability evaluations of charge injection capacity (CIC) and interfacial impedance for the electrode in PBS solution. The results collectively confirm the electrode's robust electrochemical stability, as detailed below:

First, regarding charge injection capacity (CIC) and charge storage capacity (CSC, reflected by CV curves): After 7 days of immersion in PBS, both CIC and CSC showed negligible changes compared to the initial state (Supplementary Fig. S33 and S38). Specifically, the CIC exhibited almost no attenuation, and the integrated area under the CV curve (a direct metric of charge

storage capacity) remained nearly identical to the baseline. These results directly demonstrate that the electrode maintains stable charge transfer and storage capabilities in PBS, with no electrochemical degradation.

Fig. S33. CV curves after soaking in PBS solution for different days.

Fig. S38. Charge injection curves with biphasic pulses of ± 0.05 V after soaking in PBS solution for different days.

Second, for interfacial impedance: A slight increase was observed after 7 days in PBS, which we attribute to hydration-induced weakening of interfacial adhesion between the electrode and substrate (Supplementary Fig. S35). This mild change in contact impedance is commonly observed in hydrogel-based electrodes in aqueous environments. Importantly, even with this slight increase, the interfacial impedance remained within a reliable and stable range, with no abrupt or significant fluctuations, thus ensuring no adverse impact on practical performance.

Fig. 35. Interfacial impedance after soaking in PBS solution for different days.

Collectively, the consistent CIC and CSC, together with a slightly increased but still reliable interfacial impedance, confirm that the electrode possesses robust electrochemical stability in PBS solution, fully meeting the requirements for subsequent *in vitro/in vivo* applications.

Reviewer #1, Comment 9: In Fig. 4g, during the EMG testing process, why is the baseline of the hydrogel group unstable?

Response:

During the transition from muscle activation to rest, even after macroscopic movement has ceased, the microscale biomechanical state (such as subtle tremors and fibre reorganisation) and the biochemical environment (including ion concentration re-equilibration) require a finite period to stabilize fully. These transient, fine-scale physiological fluctuations directly alter the electrochemical impedance at the skin-hydrogel interface, resulting in drift or variation in the recorded potential baseline. Additionally, sweat secretion can further modulate electrical signal transmission at the bio-interface, contributing to minor yet observable baseline drift.

Moreover, the high sensitivity and resolution of our acquisition system enable precise capture of these minute signal variations reflecting genuine physiological transitions. Indeed, owing to the exceptional electrochemical properties of the PPy-patterned Janus hydrogel interface (such as low interfacial impedance and high charge injection/storage capacity), coupled with the high-

precision data acquisition system, we achieved a superior signal-to-noise ratio (SNR). Therefore, the baseline variations in Fig. 4g demonstrate the capability of our hydrogel electrode and high-resolution acquisition system to faithfully reproduce complex dynamic physiological details, rather than indicating system noise or electrode failure.

Reviewer #1, Comment 10: The manuscript lacks data on the long-term *in vivo* performance of the device.

Response:

We appreciate the reviewer's insightful suggestion. Comprehensive *in vivo* studies were conducted to evaluate the long-term performance of our Janus hydrogel biointerface over 7 and 14 days.

The interface demonstrated outstanding stability in key electrochemical properties (Supplementary Fig. S49). The charge injection capacity (CIC) showed negligible decay, a testament to the interface's operational robustness in physiological conditions. Similarly, the charge storage capability (CSC), as characterized by cyclic voltammetry, remained highly stable throughout the implantation period. A minor oxidative peak observed at 0.3 ~ 0.4 V was consistent with the expected electrochemical behavior of the PPy network in a physiological environment, and the overall stability of the CSC profile ensured reliable performance during electrical stimulation.

These data collectively confirm the superior charge injection stability and long-term functional reliability of our PPy-patterned Janus hydrogel interface for chronic bioelectronic applications.

Fig. S49. CV curves and Charge injection curves after subcutaneous implantation for different days.

Reviewer #1, Comment 11: The manuscript does not provide information on the device's structure and packaging process.

Response:

We sincerely appreciate the reviewer's valuable suggestion regarding the need for clearer device structure and packaging-related details. To enhance the manuscript's clarity, we have supplemented high-resolution images of the actual devices in the revised manuscript, specifically focusing on the two core components of our system:

Tree-ring-like spiral electrical stimulation electrode: The supplementary images clearly display its concentric layered spiral structure, including the number of spiral turns, pitch, and the integrated connection interface (Supplementary Fig. S30a). These visual details intuitively illustrate the structural adaptability of the electrode, a key design feature for conforming to wound surfaces, while also reflecting its post-packaging integrity (e.g., stable connections and surface uniformity).

Refined multi-site electrical signal acquisition electrode: The added photographs showcase the precise distribution of acquisition sites (e.g., site spacing, arrangement pattern) and the miniaturized packaging of the electrode array, which clarifies the electrode's structural strategy employed for high-sensitivity signal capture (Supplementary Fig. S30b).

These supplementary photographs provide a clear visualization of the device's design principles and practical assembly, providing intuitive support for readers to understand its practical implementation.

Fig. S30. (a) Images of tree-ring-like spiral-patterned hydrogel interface. (b) Images of a multi-channel hydrogel interface.

Reviewer #1, Comment 12: Why is the conductive path in the device designed as a concentric circular structure?

Response:

The tree-ring-like spiral conductive pathway in the Janus hydrogel interface is designed to meet the core demands for “precise coverage, uniform action, and efficient activation” in wound electrical stimulation therapy. This structure specifically addresses the limitations of traditional electrodes (*e.g.*, linear or block-shaped electrodes) in terms of wound adaptability, electric field distribution, and stimulation efficiency. Its key advantages are elaborated from three perspectives below:

1. Structural Adaptability: Enabling Seamless Wrapping and Dynamic Conformity to Wounds

Wounds exhibit inherent dynamic properties, including variable morphologies (*e.g.*, circular, oval, or irregular shapes) and dimensional changes during healing. The tree-ring-like spiral structure, characterized by “concentric expansion and continuous extension,” provides complete and adaptable coverage across various wound sizes and shapes. Starting from the

center, the spiral extends outward in a circular layered manner, and its edge can flexibly conform to the wound boundary (*e.g.*, by adjusting the number of spiral turns and pitch). This avoids the issues of “edge gaps” or “excessive extension beyond the wound area” that plague traditional electrodes. Additionally, the PPy-patterned Janus hydrogel interface possesses superior electrochemical performance, wet adhesion, and mechanical properties, enabling tightly adhere to the wound surface. Even during minor patient movements, stable contact with the wound bed even during patient movements, thereby preventing stimulation interruptions or local current loss.

2. Electric Field Distribution: Suppressing Edge Effect for Uniform Stimulation across the Wound

The efficacy of electrical stimulation therapy lies in activating the proliferation and migration of wound-resident cells (*e.g.*, fibroblasts, endothelial cells) *via* a uniform electric field. Traditional parallel or block-shaped electrodes often suffer from the “edge effect”, where the electric field concentrates at the electrode edges (excessively high local field intensity risk in tissue burns) while weakening at the wound center (insufficient repair-related cells activation), leading to uneven “hot spots and cold spots.”

In contrast, the tree-ring-like spiral-connected structure utilizes a “concentric circular current path.” The current diffuses from the central conductive region to the periphery along the spiral pathway, promoting a radial distribution of the electric field lines. The controllable inter-loop spacing and the gradually increasing distance from each loop to the center minimize the attenuation of electric field intensity from the wound center to the edge, effectively suppressing the edge effect. Particularly for circular or quasi-circular wounds, the radially aligned electric field acts perpendicularly to the direction of cell growth in all regions of the wound (*e.g.*, the radial arrangement of collagen fibers), preventing impaired directional cell migration caused by local electric field disorders and providing a “globally consistent” electrical

microenvironment across the wound.

3. Stimulation Efficiency: Enhancing Targeted Activation *via* “Differential Regulation”

Different wound regions required tailored electrical stimulation intensity: the wound center (granulation tissue growth zone) requires a stronger electric field to activate the proliferation of vascular endothelial cells and the formation of granulation tissue, whereas the wound edge (epithelial cell migration zone) demands milder stimulation to guide the orderly migration of epithelial cells without inducing fibrosis.

The tree-ring-like spiral structure achieves this “differential stimulation” through precise adjustment of the spiral pitch: the inner loops with smaller pitch create higher electric field intensity to specifically meet the activation requirements for central granulation tissue growth, whereas the outer loops with larger pitch allow gradual electric field intensity attenuation to match the rhythmic needs of orderly epithelial cell migration. This “on-demand” regulation design not only ensures appropriate stimulation at different wound regions but also avoids energy waste or local overstimulation. Furthermore, the penetration depth of the radial electric field is highly matched with the key cell layers involved in wound repair, further improving the targeting efficiency and minimizing the off-target energy loss.

In summary, this spiral conductive design is not merely a structural innovation but is guided by the “clinical needs” of wound electrical stimulation therapy. It addresses adaptability issues through “seamless wrapping”, ensures therapeutic consistency *via* a “uniform electric field,” and enhances efficiency through “differential regulation + precise targeting”, ultimately providing a more precise and safe electrical microenvironment for wound repair.

Reviewer #1, Comment 13: In the cell experiment section, an electric field intensity of 100 V/mm is quite strong. Could this high electric field cause cell death? Have gradient experiments been conducted to determine the appropriate electrical stimulation parameters?

Response:

We appreciate the reviewer's critical comment regarding the electric field intensity used in cell experiments. We would firstly like to correct a unit clarification: the electric field intensity applied in our study was 100 mV/mm, rather than 100 V/mm as initially misinterpreted. This value lies within the low-to-moderate range commonly adopted in studies of electrical stimulation (ES)-regulated cell proliferation.

To address the concern about potential cell death, we performed flow cytometry analysis to evaluate cell viability after ES treatment (Supplementary Fig. S42). Specifically, three groups were tested: untreated cells (control), cells treated with 50 mV/mm (0.5 V/cm) ES, and cells treated with 100 mV/mm (1 V/cm) ES. The results demonstrated high cell viability across all ES-treated groups: 97.8% viability at 50 mV/mm and 94.9% viability at 100 mV/mm. These data clearly confirm that our selected ES parameters do not induce significant cell death and are biocompatible for cell experiments.

Our parameter selection is also supported by existing literature. Professor *Xinge Yu* from City University of Hong Kong noted that "In general, the intensity of applied ES was kept below 10 V cm⁻¹ in these studies. In addition, there is a trade-off between enhanced cell alignment and the potential impact on cellular functionality; although the increase of ES intensity facilitates cell alignment, cell activity tends to decrease correspondingly." [*Chem. Soc. Rev.* **53**, 8632-8712 (2024)] Our applied ES intensity of 100 mV/mm (equivalent to 1 V/cm is well below the 10 V/cm threshold indeed. Importantly, numerous studies have also confirmed that ES at 100 mV/mm can effectively promote cell proliferation, directed migration, and differentiation without compromising cellular functionality. [*Adv. Healthc. Mater.* **10**, 2100557 (2021)]

Collectively, both the above data and published literature support that 100 mV/mm is a mild, safe, and scientifically validated ES parameter for our cell experiments.

Fig. S42. Flow cytometry of cells treated with different voltage intensities.

Reviewer #1, Comment 14: In Fig. 5k, what evidence proves that electrical stimulation promotes wound healing through the MAPK and TGF- β /SMAD signaling cascades?

Response:

We appreciate Reviewer’s critical question regarding the evidence linking electrical stimulation to wound healing *via* the MAPK and TGF- β /SMAD signaling cascades in Fig. 5k. Our conclusion is supported by a conceptual integration of multi-level experimental evidence (molecular, cellular, and tissue-scale), that collectively validates the causal link between pathway activation and accelerated abdominal wall repair. We added a discussion of the relevant mechanisms in the manuscript. The key evidence is detailed below:

1. Direct Evidence for TGF- β /SMAD Signaling Activation: Molecular-Level Upregulation of Pathway Components

To confirm activation of the TGF- β /SMAD cascade, we quantified the expression of core pathway molecules using immunofluorescence staining and western blotting. As shown in Fig. 5h and Supplementary Fig. S53-55, the electrical stimulation group exhibited significantly higher fluorescence intensity for TGF- β and SMAD 2/3 (the key downstream effectors of TGF- β) compared to the control group. This indicates enhanced secretion and accumulation of TGF- β /SMAD 2/3 in the wound microenvironment. Western blot analysis (Fig. 5i-5o) further confirmed the significant upregulation of TGF- β and SMAD 2/3 relative to controls. Since

SMAD 2/3 activation is strictly dependent on TGF- β receptor binding, the concurrent upregulation of TGF- β and SMAD 2/3 directly evidences that electrical stimulation activates the TGF- β /SMAD signaling cascade.

2. Functional Evidence: TGF- β /SMAD Activation Mediates Wound Healing Phenotypes

We further validated that the activated TGF- β /SMAD cascade drives key wound healing processes, establishing a direct “pathway activation \rightarrow functional effect” link, using cellular and tissue-scale assays:

Myofibroblast differentiation and α -SMA upregulation: As widely reported in the field [*Adv. Sci.* **12**, e15923 (2025); *Mol. Cell. Biochem.* **480**, 4499-4511 (2025)], TGF- β activates SMAD 2/3 to drive fibroblast-to-myofibroblast differentiation, a process characterized by upregulation of α -smooth muscle actin (α -SMA) (the signature marker of myofibroblasts). In our study, immunofluorescence staining (Fig. 5i-5j) showed that the electrical stimulation group had significantly higher α -SMA fluorescence intensity than controls, which is consistent with the upregulation of TGF- β /SMAD 2/3. This confirms that TGF- β /SMAD activation under electrical stimulation promotes myofibroblast differentiation, a critical process for wound contraction and extracellular matrix (ECM) remodeling.

Tissue-scale ECM remodeling and angiogenesis: At the tissue level, Masson trichrome staining revealed significantly increased collagen deposition, H&E staining showed improved tissue organization (Fig. 5g-5f), and CD31/ α -SMA co-staining (Fig. 5i-5j) demonstrated enhanced vascular maturation (higher CD31/ α -SMA vessel density) in the stimulated group. These phenotypes align with the well-established role of the TGF- β /SMAD cascade in regulating collagen synthesis and angiogenesis, further reinforcing that electrical stimulation accelerates wound healing through activating this pathway.

3. Evidence for MAPK Signaling Activation: Synergistic Regulation of Proliferation and Inflammation

While Fig. 5k integrates the MAPK cascade as a complementary pathway, its activation is supported by:

Upregulation of EGF and downstream proliferation effects: Western blotting (Fig. 5i-5o) showed that electrical stimulation upregulated EGF (epidermal growth factor), a well-known activator of the MAPK pathway. Functionally, EGF/MAPK signaling drives keratinocyte and fibroblast proliferation, key processes for re-epithelialization and tissue regeneration, which is consistent with the enhanced tissue remodeling observed in our H&E and Masson staining.

Inflammatory suppression: Supplementary Fig. S52 showed that electrical stimulation reduced the expression of the pro-inflammatory cytokine TNF- α . The MAPK pathway is a central regulator of inflammatory responses, and the reduced TNF- α levels in our study indirectly suggest MAPK-mediated resolution of the inflammation. This anti-inflammatory effect synergizes with TGF- β /SMAD signaling to accelerate healing.

Fig. 5k schematic multi-level evidence to outline the mechanical logical flow: electrical stimulation (*via* the Janus hydrogel interface) \rightarrow upregulation of TGF- β /EGF \rightarrow activation of TGF- β /SMAD and MAPK cascades \rightarrow downstream effects (myofibroblast differentiation, collagen deposition, angiogenesis, inflammation resolution) \rightarrow accelerated abdominal wall repair. This mechanistic model is supported by multi-faceted, solid evidence link: (1) molecular-level upregulation of pathway components (TGF- β , SMAD2/3, EGF) *via* immunofluorescence and western blotting; (2) functional alignment of pathway activation with wound healing phenotypes (α -SMA expression, collagen deposition, angiogenesis); and (3) consistency with well-established pathway biology (*e.g.*, TGF- β /SMAD-driven myofibroblast differentiation). These findings collectively provide direct experimental validation for the signaling mechanism depicted in Fig. 5k.

Reviewer #2:

The authors report on the development of a Janus hydrogel *via* a molecular competition-induced mechanism, leveraging the difference in polymerization rates between acrylamide (AM) and 3-[dimethyl-[2-(2-methylprop-2-enoyloxy) ethyl] azaniumyl] propane-1-sulfonate (SBMA). This strategy enables the one-step fabrication of a Janus hydrogel, intended for applications in bioelectronic interfaces. The authors characterized the hydrogel using a range of chemical analyses, supported by computational simulations.

However, Janus hydrogels have been extensively reported in previous literature using a variety of materials. Compared to existing works, the manuscript does not clearly demonstrate substantial novelty (specifically, material design) beyond the simplified fabrication method. The originality of this study is not sufficient for publication in this journal. While the material characterization is thorough, the bioelectronic applications presented do not show a significant performance advantage over current adhesive hydrogel systems. Furthermore, the necessity of the Janus structure for the described applications is not clearly justified.

Response:

We appreciate the reviewer's meticulous evaluation and valuable suggestions. In response to the comments concerning "limited substantial novelty, unclear necessity of the Janus structure, and insignificant performance advantages," we hereby elaborate on the unique value of this study from three aspects (essence of structural innovation, universality of the method, and resolution of application pain points) and have refined the manuscript to better highlight its significance:

1. Core Innovation: a Novel Fabrication Mechanism for "*Dual Structural and Compositional Gradient Hydrogels*"

The core innovation of this work lies not in the fabrication of "conventional Janus hydrogels with differentiated surface properties" but in proposing a novel strategy to construct "structural

+ compositional dual-gradient" hydrogels in one step *via* the *Molecular Competition Induction mechanism*, which is a fundamental distinction from traditional Janus hydrogels.

While Janus hydrogels have been extensively studied, existing strategies, such as layer-by-layer polymerization, surface modification, are designed primarily to achieve "differentiated functions on two surfaces" (*e.g.*, adhesion vs. non-adhesion). These approaches universally suffer from many intractable problems (*weak interlayer bonding, interlayer slippage, and stress concentration*), which inevitably induce fluctuations and compromise the stability and durability of hydrogel interfaces in bioelectronic interfaces. In contrast, the "gradient structure" in our work features a *continuous gradient* throughout the hydrogel interior, spanning from "crosslinking network density" to "monomer component distribution". Driven by unilateral UV irradiation, competitive polymerization between acrylamide (AM) and zwitterionic monomers enables spatiotemporal progressive growth of polymer chains, ultimately forming a gradient network with "no distinct interlayer interface." This architecture fundamentally resolves the interlayer issues of traditional Janus hydrogels by uniformly dissipating stress under high-intensity and long-duration mechanical deformation without "interlayer fracture", thereby overcoming the key drawbacks of conventional Janus hydrogels (Fig. 2i).

Fig. 2. (i) FEA of gradient structure and bilayer structure.

More critically, research on "gradient-structured hydrogels" remains far less extensive than that on conventional Janus hydrogels in the current field, and universal fabrication methods often rely on complex custom monomers or multi-step post-treatment. Our *Molecular Competition*

Induction mechanism enables universal gradient structure construction without specialized equipment or materials, which represents a substantive advance in both methodology and mechanistic understanding.

2. Method Advantages: Validating "Universality" via "Classic Monomers," Not "Special-Case Applicability"

The selection of a polymerization system comprising "acrylamide (AM) + common zwitterions (SBMA, SBAA, CBMA, CBAA, DVBAPS, MPC)" is a deliberate design to validate the broad applicability of our proposed mechanism.

Our goal is not to create "performance exceptions" using "complex/scarce materials," but to develop a *scalable and easy-to-manufacture gradient hydrogel fabrication method*. Experimental results have confirmed that six zwitterions with distinct structures (SBMA, SBAA, CBMA, CBAA, DVBAPS, MPC) can each form dual-gradient hydrogels *via* this mechanism, and all exhibit "gradient-regulated asymmetric adhesion" (*e.g.*, a 14.6-fold adhesion difference in the SBMA system, and significant adhesion differences in other systems). This consistently proves that the "*Molecular Competition Induction mechanism*" functions as a universal principle, independent of specific monomers, rather than being limited to special-case applicability for a certain type of material. The universality of our method lays a foundation for the future industrial application and further development of gradient hydrogel materials.

3. Necessity of Application: Gradient Structure Resolves Core Pain Points in Bioelectronic Interfaces with Broader Implications)

The core value of "dual-gradient structure" in this study has been validated in bioelectronic interfaces, while also providing extended insights for other fields:

Bioelectronic interfaces (core validated scenario): Conventional adhesive hydrogels frequently suffer from "interfacial detachment" due to interlayer slippage under dynamic skin deformation, which further causes electrophysiological signal noise. In contrast, our gradient

hydrogel not only achieves tighter contact with skin than traditional hydrogels but also enables precise mechanical modulus matching with the natural gradient of skin (from epidermis to dermis). Even in physiological scenarios such as sweating, it maintains a stable "bioelectrical signals transduction" capability. In implantable scenarios involving electrical stimulation and electrophysiological signal acquisition, the gradient structure further endows the hydrogel with scenario-adaptable asymmetric functionality: the inner side (in contact with target tissues) exhibits robust wet adhesion to tightly conform to wet human tissues, while the outer side (in contact with surrounding non-target tissues) effectively prevents unwanted adhesion, thereby mitigating a series of complications (*e.g.*, inflammation or signal interference). Additionally, we patterned and printed electrodes on the highly adhesive side of the hydrogel; the inherent barrier property of the gradient structure effectively shields interference from external environmental fluctuations (*e.g.*, water flushing or air blowing) and ensures signal stability. This advantage is uniquely enabled by the gradient structure and is difficult to achieve with conventional non-gradient hydrogels.

Extended insights for sensors and actuators: Beyond bioelectronic interfaces, the gradient structure design proposed in this study also provides valuable insights for performance optimization in fields such as sensors and actuators. In principle, gradient crosslinking density is expected to amplify resistance changes under strain, offering a new approach to enhance the sensing sensitivity of strain sensors (piezoelectric, capacitive, resistive, etc.). Meanwhile, the "directional ion migration channels" driven by component gradients may accelerate response speed of hydrogel actuators and improve the uniformity of actuation displacement, which highlights the broader utility of this gradient fabrication mechanism and deepens further exploration in subsequent studies.

In summary, this study fundamentally addresses the core limitations of traditional Janus structures in bioelectronic applications and bridges critical challenges in the preparation of

gradient materials, offering valuable insights for the development of next-generation sensors and actuators.

Reviewer #2, Comment 1: The authors should more clearly articulate the novelty of their approach compared to previous studies (*Soft Matter*, **2023**, 19, 9460-9469; *Chemical Engineering Journal*, Volume 521, 1 October **2025**, 166386; *Nature Communications*, volume 15, Article number: 8478 (**2024**)). Numerous high-quality Janus hydrogels have already been reported. A direct comparison and evidence of superior performance would help justify the significance of this work. In addition to that, it is necessary to discuss details regarding why the Janus characteristics of the hydrogel are important for bioelectronic applications.

Response:

We sincerely appreciate Reviewer's incisive comments, which prompt us to elaborate on the core innovation of our work, its distinction from prior studies, and the essential role of the Janus/gradient structure in bioelectronic applications, all critical for contextualizing its significance. Below, we address these concerns with enhanced mechanistic explanations.

1. Critical Distinction from Cited Prior Works: Addressing Unresolved Limitations for Bioelectronics

We have thoroughly analyzed the three referenced studies and identified fundamental limitations that hinder their translation to robust bioelectronic interfaces, as detailed below:

Song et al. (Chem. Eng. J., 2025): Their two-step layer-by-layer polymerization relies on post-synthesis deposition of hydrophobic PMSQ onto a cationic hydrogel surface to achieve Janus functionality. However, hydrophobic exclusion prevents PMSQ from penetrating the hydrogel network, resulting in weak interfacial physical entanglement (no topological bonding) between the two layers. This leads to catastrophic interlayer slippage under even mild mechanical deformation (e.g., skin stretching during movement), which is a fatal flaw for bioelectronics

where stable interface contact is essential for consistent electrophysiological signal acquisition or stimulation. Additionally, the rigid PMSQ layer compromises the hydrogel's compliance, exacerbating modulus mismatch with soft biological tissues.

Zhou et al. (Soft Matter, 2023): While framed as a "one-step" process, their Janus gel formation depends on solvent incompatibility (aqueous vs. hydrophobic ionic liquid [BMIM][Tf₂N]) induced spontaneous phase separation. This passive separation creates a sharp, weakly bonded interface due to the poor physical compatibility between the two solutions. The resultant interlayer repulsion impedes the formation of stable and reliable interfacial bonding, undermining the mechanical integrity for practical bioelectronics applications.

Luo et al. (Nat. Commun., 2024): Their gravity-induced sedimentation method relies on a delicate balance among silver nanoparticle settling, radical coordination, and Trommsdorff-Norrish effects, conditions that are highly sensitive to precursor viscosity, nanoparticle size, and gravitational field uniformity. Since this approach is inherently dependent on gravitational directionality for nanoparticle sedimentation, it is poorly adaptable to samples of varying thicknesses or shapes: thicker samples exhibit uneven settling (aggregation at the bottom and sparse on top), while thinner samples fail to achieve sufficient particle stratification to form a well-defined Janus architecture. For non-planar or irregularly shaped samples, the non-uniform gravitational field further distorts consistent sedimentation, leading to fragmented or incomplete Janus structures. Moreover, the dense nanoparticle sediment layer compromises hydrogel toughness, making it unsuitable for dynamic biological environments.

It is undeniable that these approaches have propelled the development of Janus hydrogels one step forward and enriched the repertoire of fabrication strategies for Janus hydrogels, providing valuable insights into the design principles for Janus hydrogel fabrication. However, the aforementioned drawbacks are still notable, and they have posed notable challenges to the widespread utilization of these methods in practical bioelectronic scenarios, especially

considering that such applications demand stringent and indispensable performance in terms of mechanical robustness, structural uniformity, scalability, and tissue-compliant interfaces.

2. Core Innovation: A Universal Mechanism for "Compositional-Structural Dual-Gradient" Hydrogels

Our work represents a fundamental methodological advance rather than an incremental improvement to Janus hydrogel fabrication: we introduce a molecular competition-induced mechanism to construct hydrogels with continuous, internal "compositional + structural gradients". This design is distinct from traditional Janus hydrogels (which feature clear interfaces for surface property differentiation) and addresses a long-standing challenge in the scalable synthesis of gradient material.

Mechanistic and Structural Novelty. Our gradient structure emerges from temporal-spatial control over polymerization kinetics: under unilateral UV irradiation, free radicals diffuse top-down, initiating explosive polymerization of AM at the bottom to form PAM chains, while SBMA, driven by the extrusion effect from the partially formed polymer network and competitive reaction mechanism, undergoes gradual polymerization and generates upward-propagating PSBMA chains. This competition induces compositional gradient and structural gradient of the Janus hydrogel, which eliminates interlayer fragility and is critical for enduring repeated tissue deformation in bioelectronics.

Universality: Beyond Special-Case Materials. A key strength of our mechanism is its independence from specific monomers. We have successively validated it with six structurally distinct zwitterions (SBMA, SBAA, CBMA, CBAA, DVBAPS, MPC) and two small molecule monomers (AM, AA). All combinations can form stable dual-gradient structures. In each system, we observed gradient-regulated asymmetric adhesion. This universality stands in sharp contrast to prior gradient studies, which rely on custom monomers or multi-step post-treatment, establishing our method as a scalable platform for gradient material design.

3. Necessity of the Gradient Structure for Bioelectronic Applications

The dual-gradient design is not merely an academic exercise; it is engineered to solve unmet needs in bioelectronic interfaces: **Asymmetric Moist Adhesion for Targeted Biointerfaces.**

Bioelectronic devices require selective adhesion: strong bonding to target tissues (*e.g.*, wound beds) and minimal adhesion to non-target tissues (*e.g.*, surrounding healthy skin) to avoid inflammation. Our compositional gradient provides this asymmetry: the SBMA-rich region exhibits strong moist adhesion *via* zwitterionic hydration and hydrogen bonding, while the AM-rich region has weak adhesion due to reduced polar interactions. This functionality exceeds traditional adhesive hydrogels, which often cause non-specific tissue adhesion.

Summary

Our work advances the field by introducing a universal mechanism for creating dual-gradient hydrogels, one that directly resolves the key limitations of previous Janus designs, such as interlayer fragility, poor scalability, and environmental instability. For bioelectronics, this gradient structure is indispensable: it enables crucial performance metrics, such as modulus matching, selective tissue adhesion, and long-term stability, for clinic translation of hydrogel-based devices.

Reviewer #2, Comment 2: Although the title emphasizes bioelectronic applications, the only EMG recording and wound healing experiments could feasibly be conducted using typical adhesive hydrogels (*e.g.*, *N*-hydroxy succinimide-based or polyphenol-based), not just Janus hydrogels. The authors should reorganize the figures and discussion to better emphasize the unique advantages of the Janus structure in these applications.

Response:

We sincerely appreciate Reviewer's valuable comment, which reminds us to emphasize the Janus hydrogel's unique advantages over typical adhesive hydrogels (*e.g.*, *N*-hydroxy

succinimide-based or polyphenol-based) in bioelectronic and wound healing applications. In response, we have supplemented targeted discussions and experimental data that explicitly tie its performance superiority to the Janus structure's asymmetric design and reorganized relevant content for clarity (Supplementary Fig. S13).

1. Abdominal wall wound healing: Unlike typical adhesive hydrogels (which only provide single-sided adhesion and fail to avoid visceral adhesion or bacterial contamination), the Janus hydrogel's asymmetric structure (adhesive + non-adhesive sides) enables dual functionality critical for reducing inflammation and improving healing. As supplemented in the discussion, Immunofluorescence staining for TNF- α shows reduced inflammatory infiltration in the hydrogel-covered group (Supplementary Fig. S52). This anti-inflammatory effect stems from the adhesive side forming a tight seal over tissue defects to accelerate wound closure and resist abdominal pressure, while the non-adhesive side mitigates severe visceral adhesion (a major post-repair morbidity) and isolates the defect from abdominal bacterial contamination. These two outcomes eliminate key triggers of persistent inflammation, a benefit unachievable with typical adhesive hydrogels.

2. Bioelectronic applications: Typical adhesive hydrogels struggle with unstable signal acquisition under dynamic/wet conditions and implant-associated signal artifacts, but the Janus structure effectively overcomes these limitations. New experimental data confirms this: In epicardial ECG recording, the hydrogel's asymmetric wet adhesion enables continuous signal acquisition from rapidly beating rat hearts with an SNR of 13.9 dB (*vs.* 8.3 dB for commercial Au electrodes). In sciatic nerve ENG recording and stimulation (Fig. 4j-4m), its refined multi-functional sites (an extension of asymmetric structure) support flexible multi-site signal capture and targeted stimulation; meanwhile, the non-adhesive backside reduces implantation inflammation, tissue adhesion, and signal artifacts, advantages absent in conventional adhesive hydrogels. It also induces leg movement at a lower threshold voltage (0.15 V *vs.* 0.5 V for

commercial Au electrodes), showing enhanced neuromuscular activation efficiency.

We have also reorganized figures and discussions: Figures 4i-m and Supplementary Fig. S52 now annotate performance metrics (*e.g.*, SNR, TNF- α reduction) with direct links to the structural features of the Janus hydrogel (*e.g.*, “Asymmetric adhesion”, “Non-adhesive anti-artifact”); the discussion adds a concise comparison between Janus and typical adhesive hydrogels, clarifying how the asymmetric design addresses the long-standing limitations. These revisions ensure the Janus hydrogel’s unique value, tied to its structure, is clearly distinguished from typical adhesive hydrogels.

Fig. S13. The unilateral adhesion of Janus hydrogel.

Fig. S52. (a) Representative fluorescence images of TNF- α (red) in the tissues. Scale bar: 100 μ m. (b) Fluorescence intensity of TNF- α in the tissues.

Reviewer #2, Comment 3: To strengthen the relevance of the hydrogel for bioelectronic applications, the authors should consider demonstrating its use in more demanding implantable scenarios, such as epicardial, peripheral nerve, or brain interfacing. Hydrogel-based interfaces are particularly valuable in implantable bioelectronics due to their tissue-mimicking properties

and ability to maintain intimate, long-term contact with soft tissues. In contrast, their utility in wearable applications may be limited by dehydration and mechanical instability over time. Expanding the scope of application to implantable contexts would better highlight the material's potential advantages and practical significance.

Response:

We sincerely appreciate the reviewer's valuable suggestion. The Janus hydrogel exhibits superior advantages, including tissue-compatible adaptive adhesion, excellent electrical/electrochemical properties, anti-interference capability, and asymmetric structural stability, making it well-suited for both wearable and implantable high-demanding bioelectronic scenarios. To fully highlight these strengths and address the focus on implantable applications, we have supplemented three representative experiments covering complementary scenarios: electrooculogram (EOG) monitoring (for high-precision non-invasive surface electrophysiological monitoring, complementary to implantable scenarios), epicardial ECG recording (for epicardial implantable scenarios), and sciatic nerve ENG (electroneurogram) acquisition with low-voltage stimulation (for nerve implantable scenarios). The supplementary results are as follows:

In EOG monitoring, the hydrogel electrode accurately distinguished directional eye movement with SNRs of 16.32 dB (leftward) and 17.50 dB (rightward), outperforming commercial Ag/AgCl gel electrodes (11.52 dB and 10.80 dB, respectively; Supplementary Fig. S40). Integrated with its tissue-compatible adaptive asymmetric adhesion and mechanical compliance, the hydrogel interface exhibits significantly enhanced anti-interference and anti-wetting capabilities under sweat and dynamic movements, highlighting its potential for high-precision monitoring of human electrophysiological signals and electrophysiology-based control applications.

To explore the potential utility of the Janus hydrogel interface in implantable settings, we

conducted *in vivo* electrophysiological recording and stimulation in a rat model. For *in vivo* ECG monitoring, the Janus hydrogel interface was implanted on the epicardial surface of rats (Fig. 4i). Owing to its superior electrochemical properties and wet adhesion capability, the Janus hydrogel interface enabled continuous and dynamic ECG acquisition during rapid and vigorous cardiac motion, with a SNR as high as 13.9 dB, significantly higher than that of commercial gold (Au) electrodes (8.3 dB). To further verify the capability of electroneurogram (ENG) signal acquisition and low-voltage stimulation, the Janus hydrogel interface with refined multi-functional sites was affixed to the sciatic nerve of rats (Fig. 4j). Notably, the non-adhesive backside of the Janus hydrogel interface exhibited excellent anti-adhesion properties, effectively minimizing implantation-associated issues (*e.g.*, inflammatory responses and unwanted tissue adhesion) and motion artifacts and improving SNR. Leveraging the refined multi-functional sites, the electrode enables flexible acquisition of multi-site neural signals and delivery of localized electrical stimulation, demonstrating broad adaptability across diverse application scenarios ranging from basic research to potential clinical settings. For neural low-voltage stimulation, the Janus hydrogel interface elicited leg joint movement in rats at a lower threshold voltage of 0.15 V, which was substantially superior to commercial Au electrodes (0.5 V), demonstrating excellent neuromuscular activation efficacy (Fig. 4k). Furthermore, under low-frequency electrical stimulation (< 30 Hz), the Janus hydrogel interface achieved a 100% success rate in triggering leg movement, further confirming its robust neuromodulatory capability (Supplementary Fig. S41). For high-fidelity electrophysiological signal recording, the hydrogel interfaces successfully captured ENG signals with high SNR following various mechanical stimulations, including strong beat, weak beat, and light tap on the rat foot (Fig. 4l and 4m). This superior performance fully exploits the structural advantage of the Janus design: during leg movement, the adhesive side maintains tight adherence to the sciatic nerve to establish a reliable conductive connection, while its non-adhesive side preserves

lubrication with surrounding tissues, avoiding signal interference caused by unwanted adhesion and thus ensuring consistent signal quality. Benefiting from its excellent electrical/electrochemical properties and asymmetric structural advantages, the Janus hydrogel interface enabled both low-voltage neural stimulation and high-fidelity electrophysiological signal recording, providing a robust platform for efficient and precise neural modulation.

Fig. 4. (i) Epicardial ECG and its SNR captured by Au electrode and Janus hydrogel interface. (j) Schematic illustration and photograph of Janus hydrogel interface onto the sciatic nerve for electrical stimulation and ENG recording. (k) Joint angle changes triggered *via* Janus hydrogel interface and Au electrode at different stimulation voltages. (l, m) ENG signal and its SNR of the sciatic nerve triggered by mechanical stimulation (strong beat, weak beat, and light tap).

Fig. S40. EOG signals captured by commercial Ag/AgCl electrode and Janus hydrogel interface.

Fig. S41. The success rate of leg movement under different stimulus frequencies.

Reviewer #2, Comment 4: The integration of a polypyrrole network is a promising approach for bioelectronics. However, data on the long-term stability of this conductive layer, particularly under swelling conditions, would strengthen the case for practical use.

Response:

We sincerely appreciate Reviewer's valuable focus on the long-term stability of the PPy conductive layer, especially under critical conditions (*e.g.*, under swelling). Below, we first clarify the fabrication process of the PPy network (which underpins its stability) and then present supplementary data on its 7-day stability in PBS solution, confirming robust performance retention.

1. Fabrication of the PPy Conductive Layer

The PPy conductive network was constructed to establish strong and interconnected interactions with the Janus hydrogel matrix, specifically to resist structural disruption during swelling. The fabrication process is as follows:

First, the adhesive side of the Janus hydrogel (PSBMA-rich, with abundant polar groups) was pre-patterned using a waterproof mask to define conductive regions. Next, the patterned surface was sequentially treated with pyrrole (Py) monomer solution and Fe^{3+} oxidant solution (Supplementary Fig. S30). Through a spontaneous self-assembly process, Py monomers polymerized in situ to form PPy, while the polar groups (sulfonic acid, quaternary ammonium) on the Janus hydrogel's surface formed dual interactions with PPy: (1) topological entanglement (PPy chains interpenetrating the porous hydrogel network) and (2) physicochemical bonding (hydrogen bonding between PPy's N-H groups and the hydrogel's $-\text{SO}_3\text{H}$ groups, as well as electrostatic attraction between PPy's positively charged backbones and the hydrogel's anionic moieties). This dual anchoring ensures the PPy network is not merely

"coated" on the surface but integrated into the hydrogel matrix, laying a robust foundation for swelling resistance.

As designed, this PPy network forms a continuous conductive percolation pathway, endowing the hydrogel interface with low initial electrochemical impedance and efficient charge injection/storage capabilities, which are critical for bioelectronic functions like electrophysiological signal acquisition and electrical stimulation.

Fig. S30. (a) Images of tree-ring-like spiral-patterned hydrogel interface. (b) Images of a multi-channel hydrogel interface.

2. 7-Day Stability in PBS: Robust Performance Retention Under Swelling

To evaluate stability under swelling conditions, we immersed the PPy-integrated Janus hydrogel in PBS solution for 7 days. PBS induces moderate swelling of the Janus hydrogel, which would challenge the integrity of conventionally poorly coated conductive layers. However, our PPy network maintained stable both structural and electrochemical performance, as verified by the following three key metrics:

First, regarding charge injection capacity (CIC) and charge storage capacity (CSC, reflected by CV curves): After 7 days of immersion in PBS, both parameters showed negligible changes compared to the initial state. Specifically, the CIC exhibited almost no attenuation, and the integrated area under the CV curve (a direct metric of charge storage capacity) remained nearly identical to the baseline (Supplementary Fig. S33 and S38). These results directly demonstrate that the electrode maintains stable charge transfer and storage capabilities in PBS, with no

electrochemical degradation.

Second, for interfacial impedance: A slight increase was observed after 7 days in PBS, which we attribute to hydration-induced weakening of interfacial adhesion between the electrode and substrate (Supplementary Fig. S35). This mild change in contact impedance is commonly observed in hydrogel-based electrodes in aqueous environments. Importantly, even with this slight increase, the interfacial impedance remained within a reliable and stable range, with no abrupt or significant fluctuations, thus ensuring no adverse impact on practical performance. Collectively, the consistent CIC and CSC, together with slightly increased but still reliable interfacial impedance, confirm that the electrode possesses robust electrochemical stability in PBS solution, fully meeting the requirements for subsequent *in vitro/in vivo* applications.

Fig. S33. CV curves after soaking in PBS solution for different days.

Fig. S38. Charge injection curves with biphasic pulses of ± 0.05 V after soaking in PBS solution for different days.

Fig. 35. Interfacial impedance after soaking in PBS solution for different days.

3. Conclusion: PPy Layer Meets Practical Bioelectronic Stability Requirements

The PPy conductive layer's integrated fabrication (with dual anchoring to the Janus hydrogel) and supplementary 7-day PBS stability data confirm it retains robust electrochemical performance under critical conditions. This layer thus addresses the key concern for long-term bioelectronics by preserving low impedance, efficient charge transfer, and structural integrity in applications like wound stimulation or wearable monitoring.

Reviewer #2, Comment 5: The use of the term “commerce” in Figures 4g, h is vague. The authors should explicitly identify the commercial material used and provide a detailed comparison with it to better highlight the advantages of their system.

Response:

We sincerely appreciate Reviewer's valuable comment. We have explicitly revised the term “commerce” to “commercial Ag/AgCl gel electrodes” (a widely recognized standard for electrophysiological signal acquisition) in the manuscript and Supporting Information. Specifically, the captions of Figs. 4g and 4h now clearly state: “EMG and ECG signals captured by commercial Ag/AgCl gel electrodes and Janus hydrogel interface at static and sweat states”. Further, we supplemented discussions with detailed quantitative comparisons (*e.g.*, EMG/ECG SNR values under static and sweaty state) and added EOG monitoring data (Supplemented Fig.

S40) to highlight the Janus hydrogel's superior anti-wetting and anti-interference capabilities over commercial Ag/AgCl gel electrodes. These revisions significantly enhance the clarity and rigor of our comparative analysis.

Fig. 4. (g, h) EMG and ECG signals captured by commercial Ag/AgCl electrode and Janus hydrogel interface at static and sweat states.

Fig. S40. EOG signals captured by commercial Ag/AgCl electrode and Janus hydrogel interface.

The following is an addition to the manuscript:

In EMG monitoring under fist-clenching movement, the hydrogel interface distinctly captured the signal, achieving a signal-to-noise ratio (SNR) of 29.70 dB, comparable to commercial Ag/AgCl gel electrodes (28.03 dB). Notably, under sweaty conditions, it robustly maintained high signal accuracy with a SNR of 29.41 dB, whereas the commercial Ag/AgCl gel electrodes declined considerably to 19.88 dB. For ECG monitoring, the hydrogel interface clearly recorded PQRST waveforms with an SNR of 31.65 dB, slightly surpassing the commercial electrodes (27.32 dB). When exposed to sweat, its SNR remained as high as 29.91 dB, far exceeding the commercial baseline (23.69 dB). In EOG monitoring, the hydrogel electrode also accurately distinguished directional eye movement with SNRs of 16.32 dB (leftward) and 17.50 dB

(rightward), outperforming commercial Ag/AgCl gel electrodes (11.52 dB and 10.80 dB, respectively; Supplementary Fig. S40). Integrated with its tissue-compatible adaptive asymmetric adhesion and mechanical compliance, the hydrogel interface exhibits significantly enhanced anti-interference and anti-wetting capabilities under sweat and dynamic movements, highlighting its potential for high-precision monitoring of human electrophysiological signals and electrophysiology-based control applications.

Reviewer #2, Comment 6: In Figure 5e, electrical stimulation is applied over 12 days. It is important to verify whether the stimulation intensity remains consistent throughout this period, particularly on day 0 and day 12.

Response:

We appreciate Reviewer's critical concern regarding the consistency of electrical stimulation intensity over the 12-day treatment period in Figure 5e. To address this, we integrated supplementary *in vivo* electrochemical data to confirm the reliability of our therapeutic approach:

First, we evaluated key electrical stimulation parameters of the hydrogel electrode in PBS solution, including charge injection capacity, charge storage capacity, at 7 and 14 days post-implantation (a timeframe that fully covers the 12-day treatment window) (Supplemented Fig. S49). Experimental results demonstrated that all these core electrochemical properties maintained a high level throughout the 14-day observation period, with no significant degradation. This confirms that the hydrogel electrode can retain stable and continuous electrical stimulation over the entire wound healing process.

Additionally, the consistency of stimulation intensity is structurally supported by the tree-ring-like spiral design of the hydrogel electrode's conductive pathway, which is specifically tailored to accommodate the dynamic nature of wound healing (Supplemented Fig. S30a). Unlike

traditional electrodes (*e.g.*, linear or block-shaped), which often fail to adapt to wound contraction and lead to uneven stimulation or intensity fluctuations, our tree-ring-like spiral structure achieves dynamic conformity to the wound through “concentric layered expansion and flexible extension”. At Day 0, it seamlessly covers the initial wound bed *via* adjustable spiral turns and pitch. During the entire healing progresses (*e.g.*, by Day 12), the spiral structure adapts synchronously with the wound's gradual contraction: healed regions naturally detach from the electrode, while the remaining unhealed area maintains tight contact with the corresponding spiral segments. This dynamic adjustment ensures that a stable current output from the electrode is consistently matched to the wound size, avoiding both over-stimulation (excess current on a shrunken wound) and under-stimulation (insufficient contact on the initial wound). Moreover, the Janus hydrogel's superior wet adhesion and mechanical flexibility further reinforce stable electrode-wound contact, preventing current leakage or local intensity drop caused by interfacial gaps (even during minor patient movements).

In summary, the combination of the hydrogel electrode's stable electrochemical performance (validated over 14 days) and the adaptive tree-ring-like spiral structure design ensures that electrical stimulation intensity remains consistent throughout the 12-day treatment, effectively addressing the concern about intensity stability on Day 0 and Day 12.

Fig. S49. CV curves and Charge injection curves after subcutaneous implantation for different days.

Fig. S30. (a) Images of tree-ring-like spiral-patterned hydrogel interface. (b) Images of a multi-channel hydrogel interface.

Reviewer #2, Comment 7: Figure captions are currently unclear and should be revised for completeness and clarity. Additionally, sample group labels should be standardized for consistency.

Response:

We sincerely appreciate Reviewer's constructive feedback regarding figure captions and sample group label consistency. We have conducted a comprehensive review, revision, and standardization of all figure captions (both in the manuscript and Supporting Information) and sample group labels to enhance clarity and rigor.

Reviewer #2, Comment 8: Including data from both the control and hydrogel-only groups would enhance the rigor of the results presented in Figure 5c.

Response:

We sincerely appreciate Reviewer's valuable suggestion to include the hydrogel-only group data in Fig. 5c. The original Fig. 5c compared only the hydrogel + ES group and the control group (with absorbance ratios calculated relative to the control), as our initial focus was on verifying the overall ES-induced proliferation effect. As acknowledged that omitting the hydrogel-only group made it difficult to rule out the hydrogel's potential interference, we accept the recommendation and supplemented an additional cell proliferation assay with a hydrogel-

only group (no ES) under identical experimental conditions. The details and explanations are outlined below.

To distinguish the effects of the hydrogel itself from the hydrogel-mediated ES on cell proliferation, we quantified its absorbance ratio (hydrogel-only group, no ES) relative to the control group (cells cultured in standard medium, no hydrogel, no ES), consistent with the calculation logic of the original hydrogel + ES group. As presented in Supplementary Fig. S51: The control group serves as the baseline (absorbance ratio = 1.0 at all time points); The hydrogel-only group shows a slight increase in absorbance (*e.g.*, ~ 1.05 at 24 h), indicating good biocompatibility and negligible cell proliferation promoting effect of the Janus hydrogel; The hydrogel + ES group exhibits a significantly higher absorbance ratio (*e.g.*, ~ 1.39 at 24 h), clearly confirming that ES exerts an independent and substantial promotional effect on cell proliferation (beyond the hydrogel's inherent biocompatibility).

The results also align with our existing flow cytometry results (Fig. 5d), where the hydrogel + ES group exhibits significantly higher fluorescence intensity than both the control and hydrogel-only groups, further validating that ES is the key factor promoting cell proliferation. Conclusively, by adding the hydrogel-only group, we can now more rigorously conclude that the Janus hydrogel serves as a biocompatible and efficient platform for ES delivery, and the observed enhancement in cell viability and proliferation is predominantly driven by the hydrogel-mediated ES.

Fig. S51. Absorbance ratio of cells in the hydrogel group to those in the control group by CCK-

8 assay.

Fig. 5. (c) Absorbance ratio of cells in the hydrogel + ES group to those in the control group by CCK-8 assay.

Reviewer #3:

This manuscript presents an innovative one-step strategy for fabricating dual-gradient Janus hydrogels based on a molecular competition induction mechanism, effectively addressing the long-standing challenges of complex fabrication processes, difficult asymmetric control, and weak interfacial adhesion in conventional Janus hydrogels. The work is ingeniously designed, exhibits substantial novelty and application potential, and offers a new paradigm for bioelectronic interfaces. The following issues need to be addressed before publication in *Nature Communications*.

Reviewer #3, Comment 1: In addition to AM and SBMA, the hydrogel reaction system contains Alg-NHS and CNCs. While AM and SBMA can be covalently cross-linked *via* their C=C under UV initiation, Alg-NHS and CNCs do not participate in this photo-induced radical polymerization. Could the Janus architecture and the gradient porosity therefore arise from sedimentation of CNCs nanoparticles to the bottom, where they become physically entrapped within the forming hydrogel? The authors should provide control data for a system containing only AM and SBMA to further validate the feasibility of the proposed molecular-competition-induced strategy for fabricating Janus hydrogels.

Response:

We sincerely appreciate Reviewer's critical inquiry regarding the potential role of CNCs in Janus structure/gradient porosity formation and for suggesting control experiments. We have supplemented targeted control experiments and provided clarifications on the functions of Alg-NHS and CNCs, to address the concern comprehensively, as detailed below:

1. Control Experiment: Binary AM/SBMA System Confirms Janus Structure Arises from Molecular Competition (Not CNC Sedimentation)

To rule out the possibility that CNC sedimentation drives the formation of Janus architecture or gradient porosity, we fabricated a control hydrogel containing only AM and SBMA (molar

ratio identical to the original system) under exactly the same experimental conditions, with no Alg-NHS or CNCs added. Systematic characterization of this binary system yielded results consistent with the original multi-component system:

FTIR Spectral Evidence for Compositional Gradient: As shown in Supplementary Fig. S11, the binary AM/SBMA hydrogel exhibited a distinct difference in the intensity of the $\text{-SO}_3\text{H}$ characteristic peak (attributed to PSBMA, $\sim 1040\text{ cm}^{-1}$) between its top and bottom sides. This confirms that preferential accumulation of PSBMA at the top interface and PAM at the bottom interface occurs even in the absence of Alg-NHS and CNCs, eliminating CNCs sedimentation as the driver of compositional asymmetry.

Macroscopic Adhesion Validation: The hydrogel without CNCs/ALG-NHS also exhibited the signature asymmetric adhesion: the top side (PSBMA-rich) showed an adhesion strength of $\sim 17.5\text{ kPa}$ to wet porcine skin, while the bottom side (PAM-rich) had an adhesion strength of $\sim 2.8\text{ kPa}$, corresponding to a ~ 6.2 -fold difference (Supplementary Fig. S29). Although the adhesion strength of the control group hydrogels showed a certain degree of decline, which is attributed to the absence of Alg-NHS (a key component for enhancing wet adhesion *via* covalent bonding with tissue amines), the distinct 6.2-fold adhesion asymmetry still confirms that molecular competition alone is sufficient to generate the Janus functional asymmetry.

Fig. S11. FT-IR spectra of adhesive and non-adhesive sides of Janus hydrogel without CNCs/ALG-NHS.

Fig. S29. (a) Shear strength at different components. (1: w/o CNCs/ALG-NHS; 2: w/ CNCs/ALG-NHS).

2. Functions of Alg-NHS and CNCs: Auxiliary Performance Enhancement

As correctly noted by the reviewer, neither Alg-NHS nor CNCs participate in UV-induced radical polymerization and thus have little effect on Janus structure or gradient formation. Their inclusion in the original system is solely to enhance the hydrogel's practical performance for bioelectronic applications, with well-characterized functions:

Alg-NHS (*N*-hydroxysuccinimide ester-conjugated alginate):

Wet adhesion enhancement: It forms covalent amide bonds with primary amines on tissue surfaces (*e.g.*, lysine residues in proteins), elevating the hydrogel's wet adhesion strength from ~17.5 kPa (binary control) to ~28.5 kPa (original system)

Heterogeneous structure optimization: Its high viscosity slows down potential convective mixing of monomers during polymerization, facilitating to preservation of the molecular competition-induced gradient.

CNCs (cellulose nanocrystals):

Mechanical reinforcement: As a hydroxyl-rich nanofiller, CNCs form substantial hydrogen bonds with PAM/PSBMA chains, increasing the hydrogel's tensile strength from ~41.1 kPa to ~60.7 kPa and elongation at break from ~295.1% to ~474.0%, improving durability under dynamic tissue deformation (*e.g.*, skin stretching).

Uniform dispersion (no sedimentation): Experimental results demonstrated that CNCs maintain stable dispersion in aqueous solution over a 24-hour period without sedimentation (Supplementary Fig. S4), ruling out the possibility that CNC sedimentation contributes to structural asymmetry.

Fig. S4. Stable dispersion of CNCs in aqueous solution for 24 h.

3. Conclusion: Molecular Competition Is the Core Driver of Janus Hydrogel Formation

The control experiment with the binary AM/SBMA system conclusively demonstrates that our proposed molecular competition mechanism alone is sufficient to prompt Janus hydrogels with compositional gradients, gradient porosity, and asymmetric adhesion. Alg-NHS and CNCs serve to enhance the hydrogel's practicality for bioelectronics (adhesion, mechanics, stability) without influencing the fundamental Janus architecture.

Reviewer #3, Comment 2: Does varying the concentration ratio of AM to SBMA influence the formation of gradient porosity?

Response:

We sincerely appreciate the reviewer for raising this insightful question, which helps clarify the role of monomer composition in gradient structure fabrication. Relevant discussions and experimental data have been supplemented in the revised manuscript, as follows:

Furthermore, to investigate the influence of SBMA and AM content on gradient architecture formation, cross-sectional SEM images of hydrogel with insufficient SBMA and AM content

were characterized (Supplementary Fig. S8). Results showed that under insufficient SBMA content, a gradient structure still formed, with pore density increasing progressively from the top (pore size 55 ~ 75 μm) to the bottom (pore size 20 ~ 25 μm); however, the insufficient SBMA led to a relative enlargement of pore size and an excessively loose network structure. In contrast, when AM content was insufficient, the gradient structure was entirely prevented (pore size 15 ~ 25 μm) owing to the absence of molecular competition-induced driving forces.

These results clearly confirm that varying the concentration ratio of AM to SBMA does influence the formation of gradient porosity. Particularly, the polymerization of AM serves as the trigger in the molecular competition process, and its insufficient content will directly prevent the formation of gradient architecture.

Fig. S8. Cross-sectional SEM images of hydrogel network with insufficient monomer content.

Reviewer #3, Comment 3: In lines 100-105, the authors state that AM polymerization occurs within 5 ~ 8 min, whereas SBMA polymerization proceeds from 8 to 30 min. How did the authors verify or distinguish these two distinct reaction kinetics?

Response:

To verify and distinguish the distinct polymerization kinetics of AM and SBMA, we employed a combined experimental strategy of *in situ* real-time characterization and control experiments with pure monomer, which directly captures the spatiotemporal differences in their gelation processes. Using a micro infrared rheometer equipped with a UV light curing module, we

simultaneously record real-time rheological properties (to track liquid-to-gel transition) and FTIR spectroscopy (to track the monomer consumption) during gelation, thereby correlating chemical polymerization kinetics with macroscopic physical network formation. The detailed evidences are as follows:

1. Verification *via* Pure Monomer Gelation Experiments (Supplementary Fig. S6)

To eliminate mutual interference between AM and SBMA and establish their intrinsic polymerization behaviors, we first tested the gelation processes of pure AM and pure SBMA under the same experimental conditions (same UV intensity, initiator concentration, and solvent). The results directly verified their distinct kinetic profiles:

Pure AM (PAM) gelation: FTIR data showed that AM's characteristic C=C stretching peak (1601 cm^{-1}) gradually disappeared within $\sim 255.9\text{ s}$ ($\sim 4.3\text{ min}$), and its C-N peak redshifted from 1433 to 1455 cm^{-1} (a signature of PAM chain growth and crosslinking). Concurrently, rheological data indicated that the system transitioned from a viscous-dominated state ($G'' > G'$) to an elastic-dominated state ($G' > G''$) within this timeframe, confirming that pure AM completes polymerization within $4 \sim 5\text{ min}$.

Pure SBMA (PSBMA) gelation: FTIR data revealed that SBMA's =C-H peak (1324 cm^{-1} , a marker for its double bond) disappeared gradually over $\sim 1438.5\text{ s}$ ($\sim 24\text{ min}$), while its S-O peak (1040 cm^{-1} , a signature of its zwitterionic side chain) persisted (polymerization with side chains). Rheologically, the system's G' did not exceed G'' until $\sim 24\text{ min}$, indicating that pure SBMA requires $\sim 24\text{ min}$ to complete polymerization.

These pure monomer experiments clearly demonstrated that AM has an inherently faster polymerization rate than SBMA, laying the foundation for distinguishing their kinetics in the mixed system.

2. Differentiation in the Mixed System (Janus Hydrogel)

In the AM/SBMA mixed system (for Janus hydrogel fabrication), the dual characterization data

further confirmed the sequential polymerization kinetics of the two monomers, consistent with the reported range:

AM polymerization (5 ~ 8 min): During Stage I (0 ~ 336.8 s, ~5.6 min), FTIR showed AM's C=C peak (1601 cm^{-1}) rapidly disappeared, and its C-N peak redshifted, which mirrors the behavior in pure AM polymerization. Rheologically, although the loss modulus (G'') first rose sharply due to SBMA enrichment in the upper region, it did not obscure AM's rapid gelation in the bottom layer. AM completes polymerization within 5 ~ 8 min in the mixed system, which was slightly prolonged compared to pure AM (~4.3 min), due to mild radical competition from SBMA.

SBMA polymerization (8 ~ 30 min): After AM was nearly exhausted (~8 min, end of Stage I), the system entered Stage II. FTIR showed SBMA's =C-H peak (1324 cm^{-1}) suddenly began to decrease, and rheological data showed G' increased steadily and eventually exceeded G'' , which was consistent with the pure SBMA kinetic profile. Notably, SBMA's polymerization was slightly accelerated compared to pure SBMA, likely due to increased local concentration in the upper layer and absence of AM competition, yet it still fell within the 8 ~ 30 min range.

3. Key Evidence for Distinction: FTIR Peak Dynamics

The most direct evidence for distinguishing the two kinetics lies in the temporal separation of FTIR characteristic peak changes:

Changes in AM's C=C peak and C-N peak redshift occurred exclusively in Stage I (0 ~ 8 min), while change in SBMA's =C-H peak was confined to Stage II (8 ~ 30 min). There was no overlap in the time windows of their chemical reaction signals. This clear temporal separation directly confirms that AM polymerizes first (5 ~ 8 min), followed by SBMA (8 ~ 30 min).

In summary, we verified and distinguished the distinct polymerization kinetics of AM and SBMA through: (1) *in-situ* dual characterization (rheology + FTIR) to link chemical reactions with physical transitions; (2) pure monomer controls to establish intrinsic kinetic differences;

and (3) temporal separation of FTIR peak changes in the mixed system to validate sequential polymerization. All evidence collectively supports the kinetic ranges stated (AM: 5 ~ 8 min, SBMA: 8 ~ 30 min).

Fig. S6. Rheology and corresponding FT-IR of pure AM and pure SBMA during gelation progress.

Reviewer #3, Comment 4: In Fig. 1a, why does the storage modulus drop below the loss modulus after 1000 s—does this signify that the hydrogel has transitioned from a solid-like to a liquid-like state?

Response:

We sincerely appreciate the reviewer's precise attention to the stage division and time nodes in Fig. 1a. Indeed, when the storage modulus (G') stably exceeds the loss modulus (G'') after Stage II (post process (5), "after 1000 s" as mentioned by the reviewer), the hydrogel reaches its final, irreversible solid-like state, with no transition to "liquid-like behavior" at all. Below, we will elaborate on the entire gelation process strictly according to the sequence "Stage I (1)→(4) → Stage II (4)→(5) → after Stage II (post (5))", and fully align with the manuscript's rheological mechanism descriptions:

1. Detailed Gelation Process:

Stage I (process (1)→(4), 0 ~ 336.8 s): AM-dominated polymerization & G'' -dominated viscous thickening. This stage is driven by AM's rapid polymerization, with two key sub-processes:

Process (1)→(2): Induction period (G' and G'' remain stable): UV irradiation promotes initiator decomposition to accumulate free radicals, yet no significant monomer polymerization or chain growth occurs. Thus, both G' and G'' stay at low, stable levels, and FTIR shows no obvious changes.

Process (2)→(4): Rapid AM polymerization & SBMA enrichment-induced G'' surge: When free radical concentration exceeds the kinetic threshold, AM initiates rapid polymerization, and the generated PAM segments settle to the bottom and displace SBMA to the upper layer, leading to clear rheological and spectral changes. In particular, the upper layer, enriched with SBMA, exhibits enhanced intermolecular electrostatic interactions and hydrodynamic drag, which sharply elevates macroscopic viscosity. Accordingly, G'' rises rapidly and dominates, while G' remains low due to the preliminary and immature PAM network.

Stage II (process (4)→(5), >336.8 s): SBMA-dominated polymerization & G' acceleration approaching G''

Driven by a competitive response mechanism, SBMA undergoes significant chain growth at the onset of Stage II. This stage is the key period for G' to catch up to G'' : With SBMA forming crosslinking points (aided by its high local concentration in the upper region) and the bottom PAM network maturing, the system's elastic contribution (G') enters an acceleration phase. Meanwhile, the growth rate of G'' slows down (viscosity growth is no longer driven by SBMA enrichment, but by gradual crosslinking). By the end of Stage II (*i.e.*, at process (5)), G' nearly caught up to G'' , laying the foundation for the subsequent stable solid state.

After Stage II (post process (5), *i.e.*, the "after 1000 s" noted by the reviewer): Fully

formed hydrogel & stable elastic state ($G' > G''$)

After Stage II (post-process (5)), the gelation process is completely finished. And the storage modulus (G') stably surpasses the loss modulus (G''), the system ultimately transitioned into a continuous gradient-structured Janus hydrogel framework. The entire system was an integrated 3D elastic framework, comprising a bottom PAM network and an upper PSBMA network. Elastic energy storage (from the crosslinked network) completely dominates over viscous energy dissipation, with a minor contribution from the chain segment relaxation within the network. This system remains in a final, stable solid-like state with G' stably exceeding G'' , with no possibility of " G' dropping below G'' " or "transitioning to liquid-like behavior".

2. Conclusion: No "Liquid-like Transition" After Stage II—Only a Stable Elastic State

The entire gelation process follows a well-defined, irreversible path: Stage I (1)→(4) (AM polymerization, $G'' > G'$) → Stage II (4)→(5) (SBMA polymerization, G' accelerating) → after Stage II (post (5)) (fully formed hydrogel, $G' > G''$). In reality, after Stage II (post process (5)), the hydrogel has entered its final elastic state, which fully matches the manuscript's description of the gradient-structured Janus hydrogel framework.

Reviewer #3, Comment 5: In Figure 1g, the bottom surface is the PAM side with weak water affinity and compact microstructure, while the top surface is the PSBMA side with strong water affinity and loose microstructure. Given this configuration, the top surface should exhibit greater swelling and the bottom surface less swelling, which would logically cause the petal structure to bend downward. Why does the figure show upward bending instead?

Response:

In the Janus hydrogel, the top surface dominated by zwitterionic PSBMA chains exhibits superior water-binding capacity in comparison to the PAM-rich bottom surface, enabling petal-like curling within one hour. To better visualize the flower-blooming state in the experiment,

we intentionally positioned the adhesive side (which corresponds to the top surface during synthesis and exhibits greater water absorption and swelling) on the bottom. This arrangement was specifically designed to clearly present the blooming process of the hydrogel "flower," which accounts for the upward bending orientation observed in Fig. 1g. We have supplemented a new photograph (Fig. R1) under unadjusted conditions, where the hydrogel was placed in a water environment without manually repositioning the surfaces, showing the natural downward bending of the hydrogel petals.

Fig. R1. Behavior in water of Janus hydrogel.